# MLLM4TS: Leveraging Vision and Multimodal Language Models for General Time-Series Analysis

## Abstract

Effective analysis of time series data presents significant challenges due to the complex temporal dependencies and cross-channel interactions in multivariate data. Inspired by the way human analysts visually inspect time series to uncover hidden patterns, we ask: *can incorporating visual representations enhance automated time-series analysis?* Recent advances in multimodal large language models have demonstrated impressive generalization and visual understanding capability, yet their application to time series remains constrained by the modality gap between continuous numerical data and discrete natural language. To bridge this gap, we introduce MLLM4TS, a novel framework that leverages multimodal large language models for general time-series analysis by integrating a dedicated vision branch. Each time-series channel is rendered as a horizontally stacked color-coded line plot in one composite image to capture spatial dependencies across channels, and a temporal-aware visual patch alignment strategy then aligns visual patches with their corresponding time segments. MLLM4TS fuses fine-grained temporal details from the numerical data with global contextual information derived from the visual representation, providing a unified foundation for multimodal time-series analysis. Extensive experiments on standard benchmarks show that MLLM4TS consistently outperforms its unimodal counterpart across both predictive (e.g., classification) and generative (e.g., anomaly detection and forecasting) tasks, ranking among the top time-series backbones. These results highlight the effectiveness of introducing visual modalities and pretrained models for robust and generalizable time-series analysis.

## 1 Introduction

Time-series analysis is a critical task across domains such as manufacturing, finance, healthcare, and environmental monitoring (Hamilton, 2020), supporting process monitoring (Xu et al., 2022), outcome prediction (Lim & Zohren, 2021), anomaly detection (Liu et al., 2024c), and broader data-driven decision-making (Mahalakshmi et al., 2016). Despite its broad utility, effective analysis remains challenging due to complex temporal dependencies, the integration of multichannel and multimodal signals, and the heterogeneous requirements of application-specific tasks, highlighting the need for a unified and generalizable framework. Motivated by the fact that

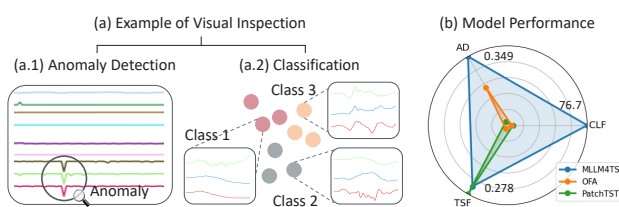

Figure 1: Illustration of (a) the effect of time series visual inspection and (b) benchmarking model performance across time-series classification, anomaly detection, and forecasting.

human analysts often rely on visual inspection to interpret time series, we consider the role of visual perception in supporting such tasks: anomalies often appear as visually salient regions (Figure 1(a)) that naturally attract attention, while classification frequently depends on characteristic motifs or recurring shapes, much like how physicians diagnose conditions by examining electrocardiogram (EKG) waveforms. These observations raise a natural question: *can we mimic human-like visual perception by integrating visual representations into time-series analysis to enhance model performance?*

Recent advances in foundation models have initiated a paradigm shift in machine learning, providing powerful general-purpose architectures capable of adapting to diverse downstream tasks (Bommasani et al., 2022). In particular, large language models (LLMs) have demonstrated strong capabilities in modeling and reasoning over sequential data (Gruver et al., 2023; Jin et al., 2024), opening new opportunities for advancing time-series analysis. At the same time, the rapid progress of large vision models has significantly improved visual representation learning (Liu et al., 2024b; Mai et al., 2025a). These developments motivate us to explore whether the capabilities of modern vision and language models can be leveraged through visual representations to improve time-series analysis.

However, existing methods for adapting LLMs to time series data often have limitations. One fundamental challenge stems from a modality mismatch: while LLMs are pretrained on discrete token sequences, time series data are inherently continuous, leading to a notable discrepancy (Gruver et al., 2023; Ni et al., 2025). Moreover, many approaches adopt patching strategies that segment time series into smaller chunks (Nie et al., 2022; Wang et al., 2024). Yet, determining an appropriate patch size is non-trivial. If the patches are too large, critical temporal information within each segment may be lost. Conversely, if the patches are too small, the model might overemphasize local features and miss the global temporal patterns in the data. Furthermore, many existing methods adopt a channel-independent (Nie et al., 2022; Zhou et al., 2023) design for multivariate time series, neglecting cross-channel dependencies that are essential for capturing their full dynamics (Wu et al., 2020; Qiu et al., 2025).

In this paper, we introduce the Multimodal Large Language Model for Time Series (MLLM4TS), a unified framework that leverages a multimodal foundation model for general time-series analysis by utilizing both time series and vision modalities. To address the limitations of language-only models, MLLM4TS introduces a vision branch that transforms multivariate time series into color-coded line-plot images, enabling the capture of global and cross-channel patterns. Visual embeddings derived from a pretrained encoder are then fused with time-series embeddings to jointly model fine-grained temporal dynamics and high-level contextual information. Extensive experimental results across time-series classification, anomaly detection, and forecasting demonstrate the effectiveness of the proposed MLLM4TS, as illustrated in Figure 1(b). In particular, MLLM4TS consistently outperforms its unimodal counterpart OFA (Zhou et al., 2023) as well as the state-of-the-art transformer-based time-series backbone PatchTST (Nie et al., 2022). Our contributions are summarized as follows:

- **Modality bridging.** By introducing a vision encoder pretrained for alignment with language-based embeddings (Radford et al., 2021), MLLM4TS effectively bridges the modality gap between continuous time series and discrete language, mitigates sensitivity to patch size selection, and enhances its ability to address complex time-series tasks. Furthermore, color-coded channels preserve channel identity while enabling global cross-channel dependency modeling.
- **Temporal-visual alignment.** We introduce a temporal-aware visual patch alignment strategy that strengthens the alignment between imaged and numerical time series by exploiting the structural properties of time-series plots. To ensure scalability for high-dimensional time series, we further employ adaptive image sizing and dimensionality reduction during visualization.
- **Versatility and generalization.** The proposed framework demonstrates promising performance across mainstream time series tasks, including classification, anomaly detection, and forecasting, and exhibits robust generalization under few-shot and zero-shot learning settings.

The remainder of this paper is organized as follows: Section 2 provides an overview of related work. Section 3 describes MLLM4TS detailing its architecture and multimodal fusion approach. Section 4 presents experimental results along with extensive ablation studies to provide deeper insights into the framework's effectiveness. Finally, Section 5 concludes the paper with implications for future work.

## 2 Related Work

This section reviews prior work in time-series analysis, covering traditional methods and recent advances in LLMs, pretrained models, and multimodal learning.

**Traditional Time Series Methods.** Traditional time-series methods, such as ARIMA (Box & Jenkins, 1970), Exponential Smoothing (Hyndman & Athanasopoulos, 2018), and Matrix Profile (Yeh et al., 2016),

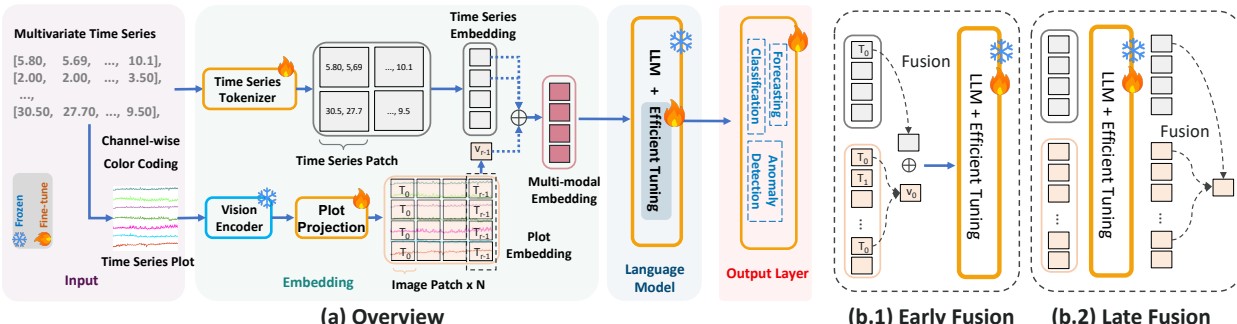

Figure 2: Overview of the MLLM4TS framework. (a) Multivariate time series are tokenized into patches and rendered as colour-coded line plots; the resulting embeddings are fused and passed to a pretrained LLM, followed by a task-specific output head. (b.1) Early fusion combines modalities before LLM processing. (b.2) Late fusion merges them after separate LLM encoding.

often struggle with multivariate or high-dimensional data due to their inherent linear assumptions. Deep learning techniques, including RNNs (LSTMs (Hochreiter & Schmidhuber, 1997), GRUs (Cho et al., 2014)), and especially Transformers (Vaswani et al., 2017), advanced time-series modeling by capturing long-range dependencies, especially in multivariate settings (Lim et al., 2021; Wen et al., 2022).

**LLMs for Time Series.** The success of LLMs in other domains (Hadi et al., 2023; Liang et al., 2024) has motivated their application to time-series analysis. However, adapting LLMs to time series remains challenging due to the modality gap between continuous signals and discrete language tokens (Liu et al., 2024e; Tan et al., 2024). Existing approaches, such as prompt engineering and patch-based tokenization (Gruver et al., 2023; Nie et al., 2022; Zhou et al., 2023), attempt to bridge this gap but still struggle to capture both global trends and local patterns, particularly in multivariate settings with complex cross-channel dependencies.

**Pretrained Time Series Models.** Pretrained time-series models have shown promise, particularly in forecasting. Approaches such as Chronos (Ansari et al., 2024) and Moirai (Woo et al., 2024) leverage large-scale data to learn general representations but remain largely restricted to forecasting, with limited effectiveness on tasks like classification and anomaly detection. Similarly, MOMENT (Goswami et al., 2024) adopts a channel-independent design that hinders the modeling of cross-channel dependencies. This task- and modality-specific orientation limits their applicability to broader time-series analysis, underscoring the need for more versatile models for general time-series analysis capable of addressing diverse tasks.

**Multimodal Time-Series Analysis.** Integrating multiple modalities, particularly visual representations, has shown promise in enhancing time-series analysis (Ni et al., 2025). Transforming time series into images, as in ViTST (Li et al., 2024), has proven effective for irregularly sampled time series classification tasks by leveraging pretrained vision transformers (Dosovitskiy et al., 2021).

Furthermore, employs a pretrained MAE He et al. (2022) for zero-shot forecasting by reformulating the task as image reconstruction. Time-VLM (Zhong et al., 2025) leverages vision–language models for few- and zero-shot forecasting, DMMV (Shen et al., 2025) uses large vision models with multi-view learning for long-term forecasting, and TAMA (Zhuang et al., 2024) applies GPT-4o for anomaly detection and interpretation. However, these methods are typically specialized for individual tasks, highlighting the need for more general multimodal architectures capable of supporting diverse time-series applications.

Despite these advancements, a unified framework leveraging visual representations across a broad range of time-series analysis tasks remains largely unexplored. This work addresses this gap by proposing MLLM4TS, a novel framework that integrates visual embeddings with LLMs to achieve robust and generalizable time-series analysis across diverse tasks, including classification, anomaly detection, and forecasting.

## 3   MLLM4TS Framework

The proposed MLLM4TS framework leverages pretrained vision and language models to capture complex temporal dependencies and enable multimodal fusion for time-series analysis. This section outlines the architecture and processing flow of MLLM4TS, as illustrated in Figure 2.

### 3.1 Model Architecture and Components

The overall MLLM4TS framework comprises four key components: the input module, embedding module, language model, and output layer. We describe each component below.

**(I) Input Module.** Given a multivariate time series $\mathbf{x}_{1:L} = \{\mathbf{x}_1, \ldots, \mathbf{x}_L\} \in \mathbb{R}^{L \times C}$ with $L$ time steps and $C$ channels, each channel is converted into a uniquely colored line plot to highlight cross-channel dependencies. These plots are horizontally stacked into a composite image (Figure 12) for the vision encoder, while the raw series is simultaneously input to the time-series tokenizer. The use of channel-wise color coding introduces additional visual contrast and texture, preventing the vision encoder (e.g., CLIP (Radford et al., 2021)) from perceiving the plotted signals as a single indistinguishable structure.

**(II) Embedding Module.** Each modality is encoded independently to produce embeddings in a shared feature space.

**Time Series Tokenizer.** The input time series is first normalized using reverse instance normalization (Kim et al., 2022), which standardizes the data and later restores it to facilitate knowledge transfer. The normalized series is then divided into non-overlapping patches (Nie et al., 2022) to capture long-range temporal dependencies with fewer tokens, and a linear projection maps each patch to the language model's embedding space for further processing.

**Vision Encoder with Plot Projection.** To model cross-channel dependencies and global patterns, each channel is transformed into a line plot, and the plots are aggregated into a composite image. A pretrained Vision-Language Model (VLM) (Zhang et al., 2024) processes this image to generate embeddings, with the visual encoder kept frozen for stability (Liu et al., 2024b;a). However, since most visual encoders are not pretrained to handle time series data, directly applying them without any adaptation may not achieve optimal performance in time-series applications. To address this, we introduce a plot projection module (a linear transformation) to adapt the visual embeddings for compatibility with the language model. This bridges the gap between channel-specific details and global information by leveraging visual data representations.

**Multimodal Embedding Fusion.** Time-series embeddings from the tokenizer and visual embeddings from the plot projection are combined to form a unified representation. To better leverage the structural information in time-series plots, we introduce a *Temporal-Aware Visual Patch Alignment* strategy (Section 3.2), enabling the model to capture complementary fine-grained temporal patterns and global cross-channel dependencies. We further explore two fusion stages: *early fusion*, which merges the two embeddings before language-model processing, and *late fusion*, which processes them separately and integrates them afterward.

**(III) Language Model.** The core of MLLM4TS consists of a pretrained language model, adapted as a pivot to process embedded multimodal data and understand sequential data through a selective fine-tuning approach. Specifically, the self-attention blocks and Feedforward Neural Network (FNN) layers are kept frozen to retain the generalized knowledge acquired during pretraining. Meanwhile, the positional embeddings and layer normalization layers are fine-tuned, allowing the model to adapt more effectively to the characteristics of time series data. This efficient tuning strategy (Mai et al., 2025b) enables adaptation to new time-series tasks with minimal task-specific data while preserving pretrained knowledge (Lu et al., 2022).

**(IV) Output Layer.** The output embeddings from the LLM are passed through a task-specific head to support a range of time series tasks, including classification, anomaly detection, and forecasting. For *classification*, a linear projection followed by a soft-max layer maps the embeddings to a probability distribution over the class set; the model is trained end-to-end with a cross-entropy loss. For *anomaly detection*, the head reconstructs the input sequence $\hat{\mathbf{x}}_{1:L} = \{\hat{\mathbf{x}}_1, \ldots, \hat{\mathbf{x}}_L\} \in \mathbb{R}^{L \times C}$, and an anomaly score is computed from the discrepancy between the original series $\mathbf{x}_{1:L}$ and its reconstruction. For *forecasting*, the head predicts the next $F$ time steps $\mathbf{x}_{L+1:L+F} = \{\mathbf{x}_{L+1}, \ldots, \mathbf{x}_{L+F}\} \in \mathbb{R}^{F \times C}$, where $F$ denotes the forecast horizon. This modular design allows MLLM4TS to flexibly adapt the same backbone to diverse downstream tasks.

### 3.2 Temporal-Aware Visual Patch Alignment

To process the 2D line-plot representation of a multivariate time series, let the image be denoted as $\mathbf{I} \in \mathbb{R}^{H \times W \times C_I}$, where $H$ and $W$ represent the image height and width, respectively, and $C_I$ denotes the

number of image color channels. The image is partitioned into non-overlapping square patches of size $P \times P$ and processed by a Vision Transformer (ViT) encoder (Radford et al., 2021). After flattening each patch, we obtain a patch sequence $\mathbf{Z}_p \in \mathbb{R}^{N \times (P^2 C_I)}$, where $N = HW/P^2$ denotes the total number of patches.

Since time-series channels are stacked vertically in the line-plot image, the horizontal axis corresponds to absolute time. Let $r = W/P$ denote the number of patches along the horizontal dimension and $q = H/P$ the number of patches along the vertical dimension. Patches that share the same horizontal index $t \in \{0, \ldots, r-1\}$ correspond to the same temporal location. We therefore group them as

$$\mathcal{S}_t = \{\mathbf{Z}_p[\,t + kr\,] \mid k = 0, \ldots, q - 1\}.$$

Applying an aggregation function (e.g., average pooling) produces a temporally aligned representation, $\mathbf{v}_t = \mathrm{Agg}(\mathcal{S}_t) \in \mathbb{R}^d$, yielding the sequence $\{\mathbf{v}_t\}_{t=0}^{r-1}$, which is aligned with the temporal ordering of the original signal as depicted in Figure 2.

In conventional vision models, spatially adjacent pixels typically encode stronger relationships. However, determining a meaningful spatial arrangement for correlated time-series channels is non-trivial and often application-dependent. In our design, the vertical axis of the line-plot image is treated as a channel index rather than a true spatial dimension. Meanwhile, the ViT encoder leverages global self-attention, which reduces reliance on local pixel neighborhoods and provides a simple yet effective mechanism for integrating information across multiple channels.

This design also removes the need for manual patch-size tuning. Specifically, the effective temporal patch length corresponds to $L/r$, where $L$ denotes the length of the time series. To ensure compatibility with the time-series branch, we further apply one-dimensional interpolation (either upsampling or downsampling) to match the temporal resolution of the time-series embeddings, thereby producing a temporally aligned multimodal representation. Empirical results demonstrate that MLLM4TS remains robust to variations in patch size (see Section 4.3).

### 3.3 MLLM4TS at Scale

To scale MLLM4TS to high-dimensional time series, we adopt an adaptive image sizing strategy. Let $C$ denote the number of channels and $p$ the number of pixels allocated per channel (we set $p = 5$). The image height and width (i.e., a square image) are defined as

$$H = \min\big(\max(C \cdot p,\ H_{\min}),\ H_{\max}\big).$$

Here, $(H_{\min}, H_{\max})$ define lower and upper bounds to ensure compatibility with downstream vision encoders (e.g., $H_{\min} = 224, H_{\max} = 420$). This formulation increases the visual resolution as the number of channels grows while preventing excessively large images.

When the number of channels becomes very large (e.g., exceeding 300), further enlarging the image becomes impractical because it substantially increases the number of visual patches processed by the ViT encoder, leading to higher computational and memory costs due to the quadratic complexity of self-attention, while also exceeding typical input resolution ranges of pretrained vision models and yielding diminishing gains in visual discriminability. To mitigate this issue, we introduce a dimension reduction strategy to reduce visual clutter while preserving the complete signal in the raw time-series branch. Specifically, we adopt a *selection*-based approach that removes redundant channels by selecting those with the lowest maximum pairwise correlation, thereby retaining a set of representative yet minimally redundant signals for visualization. The rationale behind this design is twofold. First, highly correlated channels often encode similar temporal dynamics; plotting all channels simultaneously introduces significant visual overlap without providing additional informative structure. Selecting less correlated channels, therefore, improves the diversity and interpretability of the visual representation while preserving the original signal semantics. Second, maintaining representative raw channels helps preserve global structural patterns in the visual plot, which can provide useful context for downstream visual encoders. As demonstrated in later experiments, this **global** information captured in the visualization contributes positively to model performance.

As an alternative, one could adopt an *engineering*-based approach that applies dimensionality reduction techniques (e.g., PCA (Maćkiewicz & Ratajczak, 1993)) to project the multivariate time series into a smaller

set of components used solely for visualization. Empirical results show that the *selection*-based approach consistently outperforms plotting all channels simultaneously, highlighting the effectiveness of reducing redundant visual information to avoid clutter. Moreover, it outperforms the *engineering*-based approach, suggesting that preserving representative original channels is more effective than relying on projections that may distort or obscure meaningful temporal structures (see Section 4.2).

## 4 Experimental Analysis and Discussion

We conduct a comprehensive evaluation of MLLM4TS across mainstream time-series analysis tasks to address the following research questions (RQs). The key findings are presented in this section, while additional details are provided in the Appendix B.

- ***RQ1.*** Does incorporating visual representations enhance the performance of general time-series analysis tasks (Section 4.1)?

- ***RQ2.*** What types of visual representations (e.g., image layouts, visual encoders) are most effective when integrated into the MLLM4TS framework (Section 4.2 and 4.3)?

- ***RQ3.*** Are language models actually useful for multi-modal time-series analysis (Section 4.4)?

### 4.1 Performance Overview

**A Motivating Example.** Figure 3 presents a motivating example comparing input modalities for time-series classification on five randomly sampled UEA datasets (Bagnall et al., 2018). The multi-modal approach consistently outperforms unimodal baselines, underscoring the importance of integrating local temporal and global contextual information.

**Main Results.** We evaluate MLLM4TS across three core time-series analysis tasks: classification (Section 4.1.1), anomaly detection (Section 4.1.2), and forecasting (Section 4.1.3), as well as zero-shot learning (Section 4.1.4).

For each task, we outline the experimental setup and report results to evaluate the effectiveness of the proposed framework. For fair comparison with the LLM-based TS-only framework OFA (Zhou et al., 2023), we adopt GPT-2 (Radford et al., 2019) as the language model backbone, consistent with prior sequential modeling work. For the vision encoder, we use CLIP-ViT-L-14 (Radford et al., 2021), pretrained for vision-language alignment and well-suited for

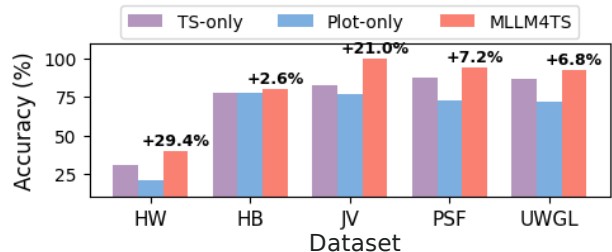

Figure 3: Performance comparison of using time series only, plot only, and the multi-modal embeddings.

visualized time-series data. Both MLLM4TS and the reproduced OFA baseline are evaluated over five random runs (error bars in Appendix B.1). Benchmark results are cited from original literature under the same protocol, or reproduced when unavailable. Full experimental details are provided in Appendix A.

#### 4.1.1 Time-Series Classification

**Settings.** For the classification task, we follow the established benchmarking protocols (Zhou et al., 2023; Wu et al., 2023; Gao et al., 2024) on UEA datasets (Bagnall et al., 2018). These datasets, as shown in Table 6, include diverse time series data across domains such as sensor readings, EEG, audio, and speech. Each dataset provides different characteristics in terms of sequence length, number of classes, and data type, offering a comprehensive evaluation in diverse classification scenarios. The model is fine-tuned using cross-entropy loss to minimize classification error.

**Results.** As shown in Figure 4, MLLM4TS outperforms these baselines, highlighting the advantages of its multimodal embedding and fine-tuning strategy. The combination of time series tokenization and vision-based

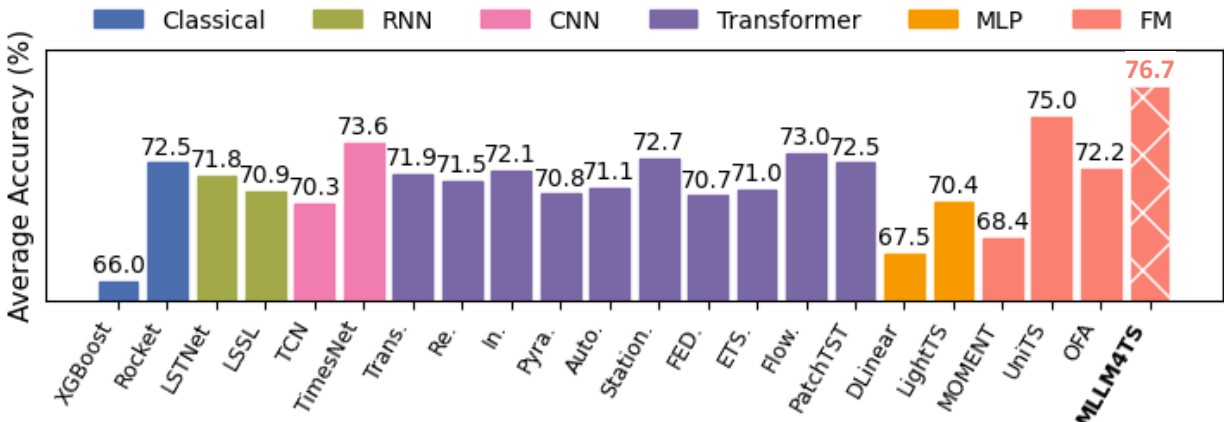

Figure 4: Model comparison in classification. "*." in the Transformers indicates the name of *former. The results are averaged from 10 subsets of UEA. See Table 9 for full results.

embeddings helps MLLM4TS capture both local temporal features and global cross-channel dependencies, resulting in improved classification accuracy across most datasets.

### 4.1.2 Time-Series Anomaly Detection

**Settings.** Anomaly detection in time series data is essential for industrial applications, including health monitoring, space exploration, and environmental monitoring. However, progress in evaluating and benchmarking anomaly detection methods has been hindered by issues related to dataset quality, such as mislabeling, bias, and feasibility concerns (Wu & Keogh, 2021; Liu & Paparrizos, 2024). To ensure reliable comparison, we conduct our evaluation using the recently published TSB-AD-M benchmark (Liu & Paparrizos, 2024), a heterogeneous and curated collection comprising 200 multivariate time series (180 for evaluation) from six time series domains. A detailed description of datasets is provided in Table 7.

In addition, to address limitations of traditional evaluation measures - specifically their susceptibility to bias, lack of discrimination and adaptability (Liu & Paparrizos, 2024), we adopt the VUS-PR (Paparrizos et al., 2022; Boniol et al., 2025) as our evaluation measure. VUS-PR improves robustness by reducing sensitivity to temporal lag, enhances accuracy by minimizing bias across different scenarios, and promotes fairness by ensuring consistent evaluations under similar conditions. Additional evaluation results based on the imperfect yet widely employed point-adjusted F score are available in Table 11. For fair comparisons with baseline methods, we use mean squared error (MSE) as the reconstruction loss across all reconstruction-based approaches. During fine-tuning, MLLM4TS is trained to accurately reconstruct normal time series patterns, with anomalies expected to result in higher reconstruction errors.

Table 1: Model comparison in anomaly detection. The top-performing models from each category in the TSB-AD-M benchmark (Liu & Paparrizos, 2024) are presented, with the detailed evaluation provided in Table 10. VUS-PR is adopted as the primary evaluation measure. Higher values indicate better performance. The best performance is highlighted in bold, and the second-best is underlined.

| Domain | Statistical | | | | | NN | | | | | | FM | |
|---|---|---|---|---|---|---|---|---|---|---|---|---|---|
| | PCA (2017) | KMeansAD (2001) | CBLOF (2003) | MCD (1999) | OCSVM (1999) | CNN (2018) | OmniAnomaly (2019) | LSTMAD (2015) | USAD (2020) | AutoEncoder (2014) | PatchTST (2022) | OFA (2023) | MLLM4TS (Ours) |
| Environment | **1.000** | 0.862 | **1.000** | **1.000** | 0.810 | 0.998 | 0.813 | 0.991 | 0.813 | 0.997 | 0.723 | 0.909 | **1.000** |
| Facility | 0.678 | 0.363 | 0.567 | 0.551 | 0.579 | 0.529 | 0.535 | 0.590 | 0.530 | 0.631 | 0.448 | 0.647 | **0.679** |
| Finance | 0.103 | 0.020 | 0.032 | 0.060 | 0.024 | 0.022 | 0.021 | 0.022 | 0.021 | 0.028 | 0.030 | **0.156** | 0.143 |
| HumanActivity | **0.278** | 0.093 | 0.137 | 0.163 | 0.113 | 0.165 | 0.197 | 0.183 | 0.197 | 0.142 | 0.103 | 0.110 | 0.122 |
| Medical | 0.113 | 0.187 | 0.073 | 0.073 | 0.070 | 0.188 | **0.300** | 0.153 | 0.278 | 0.071 | 0.138 | 0.083 | 0.131 |
| Sensor | 0.090 | **0.255** | 0.110 | 0.112 | 0.115 | 0.164 | 0.115 | 0.128 | 0.111 | 0.125 | 0.087 | 0.125 | 0.194 |
| TSB-AD-M | 0.310 | 0.295 | 0.273 | 0.271 | 0.265 | 0.313 | 0.312 | 0.307 | 0.304 | 0.295 | 0.231 | 0.296 | **0.349** |

**Results.** As shown in Table 1, MLLM4TS achieves a substantial improvement over its time-series-only counterpart, OFA, and attains the best overall performance in multivariate time series anomaly detection. It outperforms both statistical and neural network-based baselines, highlighting the effectiveness of introducing vision modality in identifying anomalies in time series.

### 4.1.3 Time-Series Forecasting

**Settings.** For multivariate time series forecasting, we follow the experimental protocol established by the recent LLM-based forecasting method AutoTimes (Liu et al., 2024e), which incorporates a diverse set of real-world datasets, including ETTh1 (Zhou et al., 2021), ECL, Traffic, Weather (Wu et al., 2021), and Solar-Energy (Liu et al., 2023). Detailed dataset descriptions are provided in Table 8. We adopt the "One-for-One" (Liu et al., 2024e) evaluation across all methods (i.e., training a separate model for each forecasting horizon). Additional discussion on auto-regressive forecasting is in Table 17. To ensure a fair comparison, we adopt GPT-2 as the backbone for all LLM-based baselines (e.g., OFA Zhou et al. (2023) and AutoTimes Liu et al. (2024e)). In addition, all baselines are evaluated under the same context length $L = 672$ to maintain consistent experimental settings. Whenever available, we report the results from the original publications; otherwise, we reimplement the baselines based on their publicly released repositories.

Table 2: Model comparison in forecasting. All the results are averaged from 4 different prediction lengths {96, 192, 336, 720}. The best performance is highlighted in bold, and the second-best is underlined. Full results are provided in Table 12.

| Models | MLLM4TS (Ours) | | OFA (2023) | | TimeVLM (2025) | | VisionTS (2025) | | AutoTimes (2024e) | | TimeLLM (2024) | | UniTime (2024d) | | iTrans. (2023) | | DLinear (2023) | | PatchTST (2022) | | TimesNet (2023) | |
|---|---|---|---|---|---|---|---|---|---|---|---|---|---|---|---|---|---|---|---|---|---|---|
| Metric | MSE | MAE | MSE | MAE | MSE | MAE | MSE | MAE | MSE | MAE | MSE | MAE | MSE | MAE | MSE | MAE | MSE | MAE | MSE | MAE | MSE | MAE |
| Weather | **0.225** | 0.266 | 0.231 | 0.269 | 0.227 | 0.266 | 0.236 | 0.269 | 0.242 | 0.278 | 0.227 | 0.266 | 0.260 | 0.283 | 0.238 | 0.272 | 0.240 | 0.300 | 0.226 | **0.264** | 0.259 | 0.287 |
| Solar. | 0.188 | 0.246 | 0.229 | 0.296 | **0.187** | 0.249 | 0.231 | 0.266 | 0.197 | **0.242** | 0.234 | 0.293 | 0.254 | 0.291 | 0.202 | 0.269 | 0.217 | 0.278 | 0.189 | 0.257 | 0.200 | 0.268 |
| ETTh1 | 0.408 | 0.430 | 0.426 | 0.438 | 0.436 | 0.452 | 0.398 | **0.415** | **0.397** | 0.425 | 0.409 | 0.432 | 0.438 | 0.445 | 0.438 | 0.450 | 0.423 | 0.437 | 0.413 | 0.431 | 0.458 | 0.450 |
| ECL | 0.165 | 0.261 | 0.167 | 0.264 | 0.168 | 0.267 | **0.157** | **0.251** | 0.173 | 0.266 | 0.170 | 0.275 | 0.194 | 0.287 | 0.161 | 0.256 | 0.177 | 0.274 | 0.159 | 0.253 | 0.192 | 0.295 |
| Traffic | 0.406 | 0.283 | 0.416 | 0.295 | 0.420 | 0.299 | 0.395 | **0.261** | 0.406 | 0.276 | 0.402 | 0.294 | 0.460 | 0.301 | **0.379** | 0.272 | 0.434 | 0.295 | 0.391 | 0.264 | 0.620 | 0.336 |
| Average | 0.278 | 0.297 | 0.294 | 0.312 | 0.288 | 0.307 | 0.283 | **0.292** | 0.283 | 0.297 | 0.288 | 0.312 | 0.321 | 0.321 | 0.284 | 0.304 | 0.298 | 0.317 | **0.276** | 0.294 | 0.346 | 0.327 |

**Results.** As shown in Table 2, lower MSE and MAE values indicate better forecasting performance. Overall, our MLLM4TS achieves competitive performance compared with existing baselines. Methods that incorporate visual information, including MLLM4TS, TimeVLM Zhong et al. (2025), and VisionTS Chen et al. (2025), consistently outperform unimodal approaches such as OFA and TimeLLM Jin et al. (2024), demonstrating the benefit of integrating visual representations for time-series forecasting.

Compared with TimeVLM, a recent multimodal vision-language model designed for time-series forecasting, MLLM4TS achieves comparable performance on datasets with relatively few channels and significantly outperforms it on datasets with a large number of channels (e.g., ECL and Traffic). This result highlights the effectiveness of our proposed channel modeling techniques in handling high-dimensional time series. Although PatchTST Nie et al. (2022) achieves slightly better forecasting performance than MLLM4TS, it performs substantially worse on other tasks, making MLLM4TS a more reliable backbone for general time-series analysis.

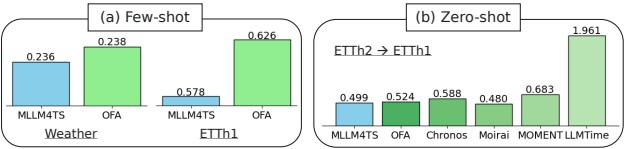

Figure 5: Performance comparison under (a) few-shot and (b) zero-shot settings. Results are reported using MSE (lower the better), averaged across four forecasting horizons {96, 192, 336, 720}. Full results are provided in Table 13 14.

### 4.1.4 Few/Zero-Shot Learning

**Settings.** LLMs have demonstrated strong few-shot and zero-shot learning capabilities in natural language tasks (Brown et al., 2020; Kojima et al., 2022). In this section, we investigate whether similar capabilities can be extended to time series analysis.

For the few-shot setting, we use only 10% of the available training data to evaluate each model's ability to adapt to data-sparse environments. For zero-shot learning, we assess the model's capacity for cross-domain generalization: specifically, we evaluate performance on a target dataset $D_A$ without any direct training on it, assuming the model has been trained or pretrained on a different source dataset $D_B$. We compare MLLM4TS with both LLM-based baselines, including OFA (Zhou et al., 2023) and LLMTime (Gruver et al., 2023), as well as recent time series foundation models, such as Chronos (Ansari et al., 2024), Moirai (Woo et al., 2024), and MOMENT (Goswami et al., 2024).

**Results.** As illustrated in Figure 5, MLLM4TS outperforms its time-series-only counterpart under both few-shot and zero-shot learning conditions. In the zero-shot scenario, it also exceeds the performance of time series foundation models that are pretrained exclusively on numerical data, thereby demonstrating superior cross-domain generalization. These findings highlight MLLM4TS's rapid adaptation to previously unseen datasets and resilience to distribution shifts.

## 4.2 Impact of Time Series Visualization

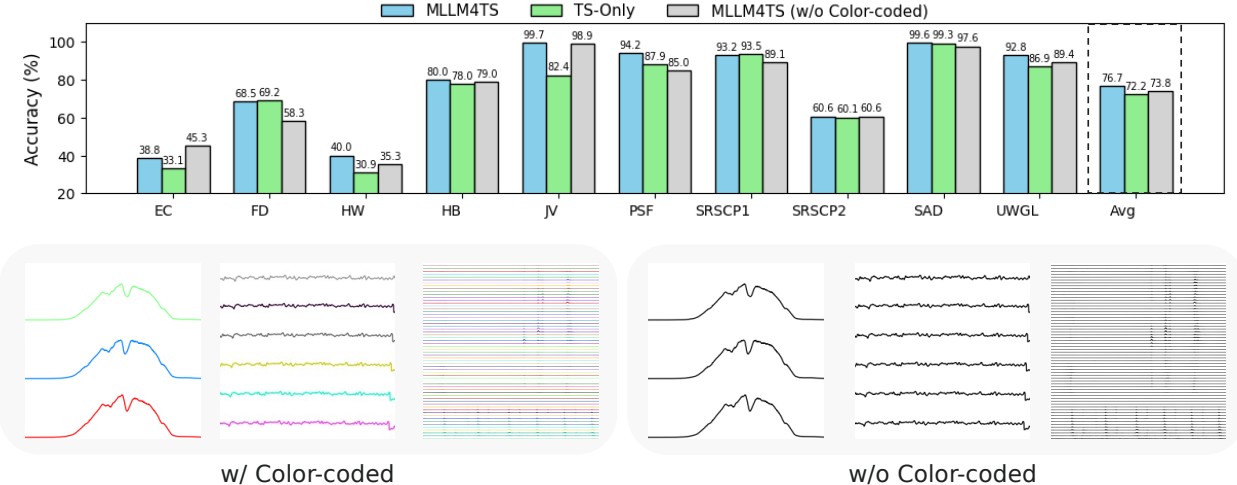

Figure 6: Impact of channel color coding on model performance.

To gain deeper insights into the role of visualization in our framework, we analyze the impact of representing time series as line plots and evaluate two design choices in our visualization pipeline: channel-wise color coding and visual dimensionality reduction.

We first investigate the effect of channel-specific **color coding** in time-series plots. As illustrated in Figure 6, we compare three configurations: (i) MLLM4TS with color-coded channels, (ii) MLLM4TS without color coding, and (iii) a time-series-only baseline. The version of MLLM4TS without color coding achieves intermediate performance between the other two settings. This observation indicates that while incorporating visual representations alone already improves performance, applying channel-wise color coding further enhances the model's ability to capture cross-channel dependencies. The added color contrast introduces richer visual textures, preventing the vision encoder from interpreting the plotted lines as a single blob.

Furthermore, we examine the effect of reducing visual dimensionality during plotting. In high-dimensional datasets such as Electricity (321 channels) and Traffic (862 channels), reducing the plotted channels to 50 using the selection-based approach consistently improves performance compared with visualizing all channels (see Table 3). In contrast, the engineering-based dimensionality reduction approach does not outperform the original visualization, likely due to information loss or signal distortion introduced by the projection process. The improvement achieved by the selection-based approach not only enhances visualization clarity and scalability but also suggests that representative subsets of channels can effectively preserve **global** structural information and inter-channel relationships. Moreover, presenting a subset of representative

Table 3: Impact of Dimensionality Reduction. MLLM4TS (Orig.) denotes visualization using all original channels. MLLM4TS (Sel.) reduces the plotted channels to 50 by removing highly correlated time-series channels, thereby retaining a representative subset of signals. MLLM4TS (Eng.) applies dimensionality reduction techniques during visualization to project the multivariate time series into a 50-dimensional representation (see details in Section 3.3).

| Method | Horizon | MLLM4TS (Orig.) | | MLLM4TS (Sel.) | | MLLM4TS (Eng.) | |
|---|---|---|---|---|---|---|---|
| | Metric | MSE | MAE | MSE | MAE | MSE | MAE |
| ECL ($C = 321$) | 96 | 0.141 | 0.244 | **0.134** | **0.232** | 0.142 | 0.243 |
| | 192 | 0.157 | 0.253 | **0.153** | **0.251** | 0.156 | 0.255 |
| | 336 | 0.168 | 0.263 | 0.169 | 0.267 | **0.163** | **0.263** |
| | 720 | **0.201** | **0.293** | 0.204 | 0.295 | 0.203 | 0.296 |
| | Avg | 0.166 | 0.263 | **0.165** | **0.261** | 0.166 | 0.264 |
| Traffic ($C = 862$) | 96 | 0.386 | 0.281 | **0.378** | **0.268** | 0.398 | 0.288 |
| | 192 | 0.399 | 0.289 | **0.396** | **0.281** | 0.411 | 0.297 |
| | 336 | 0.420 | 0.304 | **0.404** | **0.280** | 0.418 | 0.305 |
| | 720 | 0.449 | 0.308 | **0.446** | **0.304** | 0.451 | 0.310 |
| | Avg | 0.414 | 0.295 | **0.406** | **0.283** | 0.419 | 0.300 |

channels provides a clearer global view of the system dynamics, which complements the numerical time-series branch in capturing overall temporal patterns.

### 4.3 Visual Representation Analysis

Table 4: Visual representation analysis on image layouts, visual encoders, fusion strategies, and patch-size sensitivity. Accuracy is reported for classification (CLF) and VUS-PR for anomaly detection (AD). Better performance is highlighted in bold. Full results are in Tables 15 and 16.

| Task | Img Layout | | VisualEnc | | FusionStage | | PatchSize STD | |
|---|---|---|---|---|---|---|---|---|
| | Horizontal | Grid | CLIP | ResNet | Early | Late | Plot-TS | TS-Only |
| CLF | **76.7** | 75.2 | **76.7** | 72.6 | **76.7** | 73.5 | **0.56** | 1.13 |
| AD | **0.349** | 0.344 | **0.349** | 0.348 | **0.349** | 0.343 | – | – |

With the promising results achieved by our multimodal strategy across mainstream time series analysis tasks, we further investigate the role of different visual representations. This analysis is conducted from four perspectives, as illustrated in Table 4: image layout, visual encoder choice, fusion stage, and sensitivity to patch size selection. For **image layout**, we compare two configurations: the horizontal layout, where each time series channel is stacked horizontally, and the grid layout, where each channel is plotted within a smaller subregion of the image (a visual example provided in Figure 11). Overall, the horizontal layout consistently outperforms the grid layout, highlighting the effectiveness of our proposed temporal-aware visual patch alignment for aligning visual and time series modalities. To assess the impact of the **visual encoder**, we compare two representative architectures: CLIP (Radford et al., 2021), which is pretrained on vision-language alignment tasks, and ResNet (He et al., 2016), which is pretrained solely on image classification. Across tasks, CLIP consistently outperforms ResNet, underscoring its superior capability in processing visual representations of time series due to its alignment with language-based embeddings.

Moreover, we investigate the effectiveness of different **fusion strategies**, with particular focus on the stage at which modalities are integrated. Experimental results show that early fusion consistently achieves better performance, supporting the hypothesis that *low-level* correlations between imaged time series and numerical time series encode meaningful and complementary information for time-series analysis (Baltrušaitis et al., 2018; Mo & Morgado, 2024). In addition to its predictive advantages, early fusion exhibits reduced computational cost, as it minimizes the number of tokens that need to be processed by the language model. A detailed runtime comparison is presented in the following section in Figure 7.

We further investigate the impact of the vision modality on the model's **sensitivity to patch size selection**. As shown in the "PatchSize STD" column of Table 4, we report the standard deviation of classification performance across different patch sizes (ranging from $\{1, 2, ..., 10\} \times L/r$, see notation in Section 3.2). Note that for anomaly detection, patch size variation is not applicable, as each input instance corresponds to a fixed-length time series window. Compared to its TS-only counterpart, MLLM4TS exhibits lower performance variance, indicating greater robustness to patch size variation. This stability suggests that the temporal-aware visual alignment preserves the temporal structure more effectively and reduces the model's dependence on precise patch size selection. Further implementation details on patch size selection are provided in Appendix A.2.

### 4.4 Language Backbone Analysis

Table 5: Comparison of performance between the LLM and LLM2Attn backbones. "$\Delta$ Perf" denotes the relative percentage by which LLM outperforms LLM2Attn (positive: LLM better; negative: LLM2Attn better). Results are averaged over 10 UEA datasets for classification (Accuracy), TSB-AD-M for anomaly detection(VUS-PR), and the Weather dataset for forecasting (MSE). See Appendix B.3 for details.

| Task | TS-Only | | | Plot-TS (Ours) | | |
|---|---|---|---|---|---|---|
| | LLM | LLM2Attn | $\Delta$ Perf | LLM | LLM2Attn | $\Delta$ Perf |
| CLF | 72.2 | 70.2 | 2.80% | 76.7 | 71.4 | 6.90% |
| AD | 0.296 | 0.286 | 3.40% | 0.349 | 0.340 | 2.60% |
| Forecasting | 0.231 | 0.225 | -2.60% | 0.225 | 0.252 | 12.00% |

With the growing debate over the effectiveness of LLMs for time-series analysis, recent studies have reported that LLM-based methods offer limited advantages over models trained from scratch and fail to adequately capture sequential dependencies in forecasting tasks (Tan et al., 2024). In this work, we extend the scope of investigation to include classification and anomaly detection, examining whether the language modeling capabilities of LLMs are beneficial across a broader range of time series tasks.

As shown in Table 5, we follow the LLM4TS ablation protocol introduced in (Tan et al., 2024), where "LLM" refers to a model using a GPT-2 backbone, and "LLM2Attn" replaces the language model with a single multi-head attention layer (i.e., PAttn, as proposed in the study). In the time-series-only setting, we observe similar trends reported in prior work: replacing the LLM with a simpler attention mechanism results in a 2.6% improvement in forecasting. However, for classification and anomaly detection, models utilizing the full LLM outperform the LLM2Attn variant.

In multimodal scenarios, the benefits of language modeling become more pronounced and consistent, where all three tasks, forecasting, classification, and anomaly detection, benefit from the use of LLMs. This highlights the effectiveness of combining a language-aligned vision encoder with LLMs and underscores the importance of language modeling capabilities for general-purpose multi-modal time series analysis.

To better understand how the language model processes multimodal input, we visualize the attention maps of the language model backbone in Figure 9. Despite the promise of language modeling for this task, scaling to billion-parameter models (Touvron et al., 2023; Yang et al., 2024) does not consistently yield improvements over smaller architectures such as GPT-2. This suggests that GPT-2-scale models are sufficiently expressive, while larger models may introduce issues such as overfitting. Additional results on different language model backbones are provided in Appendix B.3.

### 4.5 Runtime Analysis

We further investigate alternative tuning strategies and their associated runtime implications (Mai et al., 2024). As shown in Figure 7(a), the addition of the vision processing branch leads to notable performance gains, albeit at the cost of increased computational overhead. The late fusion strategy incurs a higher runtime due to the longer token sequences passed to the LLM. Furthermore, the TuneAll variants, which involve fine-tuning all model parameters, do not yield improved performance despite their significantly higher

computational cost. We also analyze the training time under different fine-tuning configurations as depicted in Figure 7(b). MLLM4TS adopts the tuning strategy described in Section 3.1, where the "Freeze" variant keeps the pretrained vision and language backbones fixed and updates only the task-specific linear head. In contrast, "TuneVis" further fine-tunes the vision encoder. Among these variants, MLLM4TS achieves the best overall performance while maintaining relatively low training cost, demonstrating the effectiveness of its selective fine-tuning strategy.

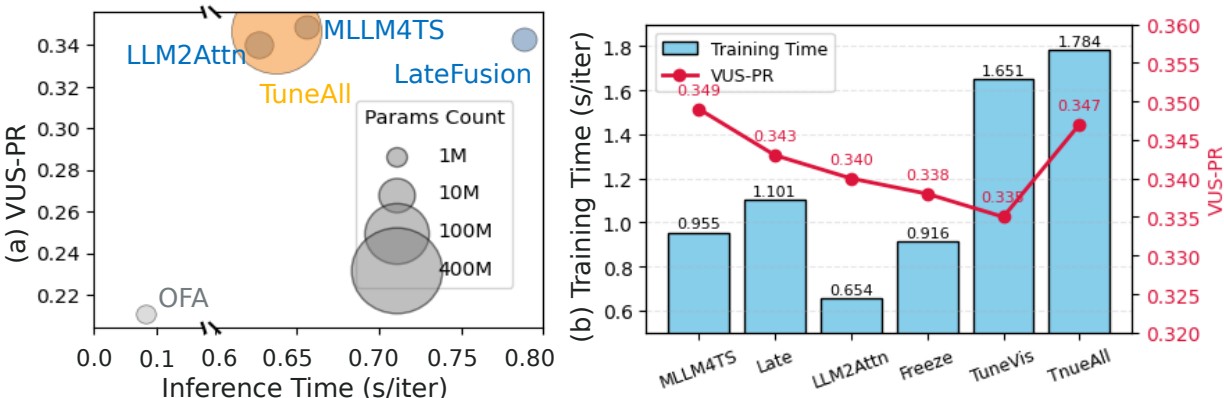

Figure 7: Efficiency evaluation of MLLM4TS and its variants on the TSB-AD benchmark (Liu & Paparrizos, 2024). (a) Inference time versus anomaly-detection performance (measured in VUS-PR (Boniol et al., 2025)), with bubble area proportional to the number of tunable parameters. (b) Training time and corresponding VUS-PR, measured using a fixed batch size of 64.

### 4.6 Discussion

We conclude this section by synthesizing research insights in response to the three research questions posed earlier. In this study, we employ a common and intuitive form of time series visualization - the time series line plot. Our findings indicate that incorporating visual representations can significantly enhance the performance of time series analysis by offering an additional modality of data representation, without requiring external information or domain-specific expert knowledge. Through systematic analysis of visual representations, we validate the benefits of exploiting structural patterns embedded in composite line plots and demonstrate the advantage of utilizing visual encoders pretrained on vision-language alignment. The superior performance of early fusion strategies highlights the presence of low-level correlations between imaged and numerical representations of time series data. Furthermore, our results underscore the importance of language modeling capabilities in multimodal time series analysis. The incorporation of a vision modality enhances performance but also introduces additional computational overhead. This observation motivates future work aimed at developing a lightweight visual encoder tailored to time series data, such as pretrained CLIP (Radford et al., 2021) for the time series domain. Overall, this work offers both a conceptual foundation and empirical evidence for the promise of multimodal large language models (Kong et al., 2025; Jiang et al., 2025), particularly in harnessing vision for advanced time series understanding (Ni et al., 2025).

## 5 Conclusion

In this paper, we introduced MLLM4TS, a unified multimodal framework for time-series analysis that leverages pretrained language models with vision-based encoders. By combining sequential and visual representations, MLLM4TS captures both local temporal patterns and global cross-channel dependencies, effectively addressing the complexities of multivariate time series. We evaluated its effectiveness on classification, anomaly detection, and forecasting across diverse datasets, demonstrating the complementary contributions of numerical and visual modalities. In summary, MLLM4TS offers a flexible solution for diverse time-series applications, opens avenues for incorporating additional aligned modalities such as images and videos, and motivates future work on lightweight encoders and more effective visual representations.

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

# Appendix

## A Experimental Setup

### A.1 Dataset Description

We evaluate MLLM4TS on standard time-series analysis tasks using widely adopted benchmark datasets. For classification and forecasting, we follow the data processing and train-validation-test split protocols established in TimesNet (Wu et al., 2023). For anomaly detection, we use the recent TSB-AD benchmark (Liu & Paparrizos, 2024; Liu et al., 2025), which addresses common concerns regarding the quality and reliability of time-series anomaly detection datasets. Detailed dataset statistics are provided in Table 6 (classification), Table 7 (anomaly detection), and Table 8 (forecasting).

Table 6: Summary of datasets used for the time-series classification task from UEA archive (Bagnall et al., 2018). The table includes the number of training and test samples, sequence length, number of time series dimensions, number of classes, and data type for each dataset. These datasets span various domains such as sensor readings, EEG, audio, and speech, providing a diverse evaluation for time-series classification.

| Dataset | Train | Test | Len | Dim | Class | Domain |
|---------|-------|------|-----|-----|-------|--------|
| EC | 261 | 263 | 1751 | 3 | 4 | SPECTRO |
| FD | 5890 | 3524 | 62 | 144 | 2 | EEG |
| HW | 150 | 850 | 152 | 3 | 26 | HAR |
| HB | 204 | 205 | 405 | 61 | 2 | AUDIO |
| JV | 270 | 370 | 29 | 12 | 9 | AUDIO |
| PSF | 267 | 173 | 144 | 963 | 7 | OTHER |
| SRSCP1 | 268 | 293 | 896 | 6 | 2 | EEG |
| SRSCP2 | 200 | 180 | 1152 | 7 | 2 | EEG |
| SAD | 6599 | 2199 | 93 | 13 | 10 | SPEECH |
| UWGL | 2238 | 2241 | 315 | 3 | 8 | HAR |

Table 7: Summary of datasets used for the time-series anomaly detection task on TSB-AD benchmark (Liu & Paparrizos, 2024). The 'Anomaly Type' column indicates whether the datasets feature point anomalies (P) or sequence anomalies (Seq).

| Domain | Dataset | # TS | Avg Dim | Avg TS Len | Avg # Anomaly | Avg Anomaly Len | Anomaly Ratio | Anomaly Type |
|--------|---------|------|---------|------------|---------------|-----------------|---------------|--------------|
| Sensor | GHL (2016) | 25 | 19 | 199001.0 | 2.2 | 1035.2 | 1.1% | Seq |
|  | Genesis (2018) | 1 | 18 | 16220.0 | 3.0 | 16.7 | 0.3% | Seq |
|  | SWaT (2016) | 2 | 59 | 207457.5 | 16.5 | 1093.6 | 12.7% | Seq |
|  | SMAP (2018) | 27 | 25 | 7855.9 | 1.3 | 196.3 | 2.9% | Seq |
|  | MSL (2018) | 16 | 55 | 3119.4 | 1.3 | 111.7 | 5.1% | Seq |
|  | GECCO (2018) | 1 | 9 | 138521.0 | 51.0 | 33.8 | 1.2% | Seq |
|  | CATSv2 (2023) | 6 | 17 | 240000.0 | 11.5 | 811.6 | 3.7% | Seq |
| HumanActivity | Daphnet (2009) | 1 | 9 | 38774.0 | 6.0 | 384.3 | 5.9% | Seq |
|  | OPP (2010) | 8 | 248 | 17426.8 | 1.4 | 394.3 | 4.1% | Seq |
| Facility | Exathlon (2021) | 27 | 21 | 60878.4 | 4.3 | 1373.3 | 9.8% | Seq |
|  | SMD (2019) | 22 | 38 | 25466.4 | 8.9 | 112.8 | 3.8% | Seq |
|  | PSM (2021) | 1 | 25 | 217624.0 | 72.0 | 338.6 | 11.2% | P&Seq |
| Finance | CreditCard (2018) | 1 | 29 | 284807.0 | 465.0 | 1.1 | 0.2% | P&Seq |
| Medical | MITDB (2000) | 13 | 2 | 336153.8 | 15.2 | 1846.8 | 2.7% | Seq |
|  | SVDB (1990) | 31 | 2 | 207122.6 | 68.3 | 268.2 | 4.8% | Seq |
|  | LTDB (2000) | 5 | 2 | 100000.0 | 105.0 | 134.4 | 15.5% | Seq |
| Environment | TAO (2006) | 13 | 3 | 10000.0 | 788.2 | 1.1 | 8.7% | P&Seq |

Table 8: Summary of datasets used for the time-series forecasting task. 'Dim' denotes the variate number. 'Dataset Size' denotes the total number of time points in (Train, Validation, Test) splits respectively. 'Forecast Length' denotes the future time points to be predicted. 'Frequency' denotes the sampling interval of time points.

| Dataset | Dim | Forecast Length | Dataset Size | Frequency | Domain |
|---|---|---|---|---|---|
| Weather (2021) | 21 | {96, 192, 336, 720} | (36792, 5271, 10540) | 10min | Weather |
| Solar-Energy (2023) | 137 | {96, 192, 336, 720} | (36601, 5161, 10417) | 10min | Energy |
| ETTh1 (2021) | 7 | {96, 192, 336, 720} | (8545, 2881, 2881) | Hourly | Electricity |
| ECL (2021) | 321 | {96, 192, 336, 720} | (18317, 2633, 5261) | Hourly | Electricity |
| Traffic (2021) | 862 | {96, 192, 336, 720} | (12185, 1757, 3509) | Hourly | Transportation |

## A.2 Implementation Details

MLLM4TS converts multivariate time series into a single composite image by plotting each channel as a color-coded line within a horizontally arranged layout. The pseudo-code for this transformation is provided in Algorithm 1, where the resulting image tensor $\hat{I} \in \mathbb{R}^{3 \times H \times W}$ is generated from the input time-series sequence $\mathbf{X} = \{\mathbf{x}_t\}_{t=1}^L \in \mathbb{R}^{L \times C}$ and subsequently used in the core model processing.

We present the core processing pipeline of MLLM4TS in Algorithm 2. The image tensor and the original time series are fed into the vision encoder and the time-series tokenization module, respectively, where the latter includes patching and linear projection. The resulting visual and temporal embeddings are aligned using the proposed temporal-aware strategy and subsequently fused before being passed to the language model backbone. The final prediction is obtained via a task-specific linear head on the last-layer hidden states as illustrated in Algorithm 3. For classification, we supervise the predicted logits $\mathbf{Y}_{\mathrm{cls}} \in \mathbb{R}^{B \times K}$ with one-hot labels $\mathbf{Y}^* \in \{0, 1\}^{B \times K}$ via the cross-entropy loss $\mathcal{L}_{\mathrm{cls}} = -\frac{1}{B} \sum_{i=1}^{B} \sum_{k=1}^{K} Y_{i,k}^* \log\big[\mathrm{softmax}(\mathbf{Y}_{\mathrm{cls},i})\big]_k$. For anomaly detection, we reconstruct the input time series $\mathbf{X} \in \mathbb{R}^{B \times L \times C}$ and minimize the mean-squared error $\mathcal{L}_{\mathrm{ad}} = \|\mathbf{Y}_{\mathrm{ad}} - \mathbf{X}\|^2$. The anomaly score is obtained via the reconstruction loss between the original and reconstructed time series. For forecasting, we predict the future series $\mathbf{X}_{L+1:L+F} \in \mathbb{R}^{B \times F \times C}$ and likewise minimize $\mathcal{L}_{\mathrm{fc}} = \|\mathbf{Y}_{\mathrm{fc}} - \mathbf{X}_{L+1:L+F}\|^2$. To handle varying numbers of input channels and enhance generalization, we adopt a cross-channel weight sharing strategy, which implicitly captures inter-variable dependencies during training (Nie et al., 2022). This mechanism complements the visual embeddings that also encode cross-channel relationships.

At the core of MLLM4TS are two pretrained backbones: the vision encoder CLIP-ViT-L-14 (Radford et al., 2021) and the language model GPT-2 (Radford et al., 2019), which are used by default unless stated otherwise. All experiments are conducted using PyTorch on NVIDIA A100 GPUs. We adopt the AdamW optimizer (Loshchilov et al., 2017) with a cosine learning rate scheduler and a warm-up starting at $10^{-6}$. Classification is trained for a maximum of 50 epochs with early stopping patience of 15, while anomaly detection and forecasting use a maximum of 10 epochs with patience set to 3. Trainable projection layers and output heads are implemented as linear layers for simplicity and efficiency. All results are averaged over five runs with different random seeds. Performance stability is illustrated in Figure 8, which includes error bars representing standard deviation.

# B Supplementary Results

In this section, we present supplementary evaluation results for MLLM4TS and baseline methods. Section B.1 provides performance variability illustrated with error bars, followed by comprehensive results for the mainstream time-series analysis tasks in Section B.2. Detailed ablation study results are reported in Section B.3.

---

**Algorithm 1** Time Series to Image

---

**Require:** Input time series $\mathbf{X} = \{\mathbf{x}_t\}_{t=1}^L \in \mathbb{R}^{L \times C}$, image size $(H, W)$
**Ensure:** Image tensor $\hat{I} \in \mathbb{R}^{3 \times H \times W}$

1:   $G \leftarrow (C, 1)$            ▷ grid rows = channels
2:   $colors \leftarrow \text{colormap}(C)$
3:   **for** $c = \{1, \ldots, C\}$ **do**
4:      $\mathbf{x}^i \leftarrow \mathbf{X}_{:,i}$            ▷ extract $i$-th channel
5:      Create subplot in row $i$ of grid $G$
6:      $\text{plot}(1 : L, \ \mathbf{x}^i)$ in color $colors[c]$
7:   **end for**
8:   Render the figure to an image tensor $\hat{I}$
9:   **return** $\hat{I}$

---

**Algorithm 2** MLLM4TS Backbone

---

**Require:** Time series $\mathbf{x}_{1:L} \in \mathbb{R}^{L \times C}$, image tensor $\hat{I} \in \mathbb{R}^{3 \times H \times W}$
**Ensure:** Fused multi-modal token features $\mathbf{F} \in \mathbb{R}^{B \times N_{\text{ts}} \times d}$, where $d$ is the feature dimension of language model

1:   **Plot Embedding:**
2:   $\mathbf{V} \leftarrow \text{VisionEncoder}(\hat{I}) \in \mathbb{R}^{B \times N_{\text{vis}} \times d_v}$           ▷ reshape $V \in \mathbb{R}^{d_v \times (B \, N_{\text{vis}})}$ if needed
3:   $\mathbf{V}' \leftarrow W_{\text{proj}} \, \mathbf{V} + b_{\text{proj}} \in \mathbb{R}^{B \times N_{\text{vis}} \times d}$           ▷ Plot Projection $W_{\text{proj}} \in \mathbb{R}^{d \times d_v}$

4:   **Time Series Embedding:**
5:   $\mu \leftarrow \text{mean}(\mathbf{x}_{1:L}, \ \text{dim} = 1) \in \mathbb{R}^{B \times 1 \times C}$
6:   $\sigma \leftarrow \sqrt{\text{var}(\mathbf{x}_{1:L}, \ \text{dim} = 1) + \epsilon} \in \mathbb{R}^{B \times 1 \times C}$
7:   $\mathbf{x} \leftarrow (\mathbf{x}_{1:L} - \mu) / \sigma$           ▷ Time Series Normalization
8:   $\tilde{\mathbf{X}} \leftarrow \text{transpose}(\mathbf{x}_{1:L}, (B, L, C) \rightarrow (B, C, L))$
9:   $\tilde{\mathbf{X}} \leftarrow \text{Padding}(\tilde{\mathbf{X}}) \in \mathbb{R}^{B \times C \times L'}$
10:   $\hat{\mathbf{X}} \leftarrow \text{Unfold}(\tilde{\mathbf{X}}, P_{ts}, S) \in \mathbb{R}^{B \times C \times N_{\text{ts}} \times P_{ts}}$           ▷ Patch size $P_{ts} = L/r$ detailed in Section 3.2
11:   $\mathbf{T} \leftarrow \text{reshape}(\hat{\mathbf{X}}, (B, N_{\text{ts}}, P_{ts} C))$
12:   $\mathbf{T}' \leftarrow W_{\text{tok}} \, \mathbf{T} + b_{\text{tok}} \in \mathbb{R}^{B \times N_{\text{ts}} \times d}$           ▷ Time Series Tokenizer $W_{\text{tok}} \in \mathbb{R}^{d \times P_{ts} C}$

13:   **Cross-modal Alignment:**
14:   $H = W = \lfloor \sqrt{N_{\text{vis}}} \rfloor$
15:   $\mathcal{V} \leftarrow \text{reshape}\big(\text{transpose}(\mathbf{V}', 1, 2), (B, d, H, W)\big)$
16:   $\bar{\mathbf{V}} \leftarrow \text{mean}_{\text{H}}(\mathcal{V}) \in \mathbb{R}^{B \times d \times W}$           ▷ Average-pool across the height (H) dimension
17:   $\widetilde{\mathbf{V}} \leftarrow \text{interp}(\bar{\mathbf{V}}, N_{\text{ts}}) \in \mathbb{R}^{B \times N_{\text{ts}} \times d}$           ▷ Linear interpolation

18:   **Fusion & decoding:**
19:   $\mathbf{Z} \leftarrow \widetilde{\mathbf{V}} + \mathbf{T}'$
20:   $\mathbf{F} \leftarrow \text{LanguageModel}(\mathbf{Z}) \in \mathbb{R}^{B \times N_{\text{ts}} \times d}$           ▷ Last hidden states
21:   **return** $\mathbf{F}$

---

---

**Algorithm 3** Task-Specific Head

---

**Require:** Final hidden states $\mathbf{F} \in \mathbb{R}^{B \times N \times d}$, mean $\boldsymbol{\mu} \in \mathbb{R}^{B \times 1 \times C}$, std. dev. $\boldsymbol{\sigma} \in \mathbb{R}^{B \times 1 \times C}$ from Algorithm 2
**Ensure:** Task outputs $\mathbf{Y}$

1: **Classification head:**
2: $\mathbf{u} \leftarrow \text{Pooling}(\mathbf{F}) \in \mathbb{R}^{B \times d}$
3: $\mathbf{u}' \leftarrow \text{LayerNorm}(\mathbf{u})$
4: $\mathbf{Y}_{\text{cls}} \leftarrow W_{\text{cls}}\,\mathbf{u}' + b_{\text{cls}} \in \mathbb{R}^{B \times K}$            $\triangleright$ Classification logits

5: **Anomaly detection head:**
6: $\mathbf{G} \leftarrow \text{LayerNorm}(\mathbf{F}) \in \mathbb{R}^{B \times L \times d}$            $\triangleright$ $L = N$ in anomaly detection
7: $\mathbf{R} \leftarrow W_{\text{ad}}\,\mathbf{G} + b_{\text{ad}} \in \mathbb{R}^{B \times L \times C}$
8: $\mathbf{Y}_{\text{ad}} \leftarrow \mathbf{R} \times \boldsymbol{\sigma} + \boldsymbol{\mu}$            $\triangleright$ Reconstructed time series

9: **Forecasting head:**
10: $\mathbf{H} \leftarrow \text{LayerNorm}(\mathbf{F}) \in \mathbb{R}^{(B \times C) \times (N \times d)}$       $\triangleright$ Cross-channel weight sharing mechanism (Nie et al., 2022)
11: $\mathbf{P} \leftarrow W_{\text{fc}}\,\mathbf{H} + b_{\text{fc}} \in \mathbb{R}^{(B \times C) \times F}$
12: $\mathbf{P}' \leftarrow \text{reshape}(\mathbf{P}, (B, C, F))$
13: $\mathbf{Y}_{\text{fc}} \leftarrow \mathbf{P}' \times \boldsymbol{\sigma} + \boldsymbol{\mu}$            $\triangleright$ Predicted time series
14: **return** $\{\mathbf{Y}_{\text{cls}}, \mathbf{Y}_{\text{ad}}, \mathbf{Y}_{\text{fc}}\}$

---

## B.1 Error Bars

We report the performance standard deviation of MLLM4TS across five random seeds in Figure 8, based on four evaluation measures available in the TSB-AD benchmark (Liu & Paparrizos, 2024). The consistently low standard deviation across all metrics suggests that MLLM4TS exhibits stable and reliable performance.

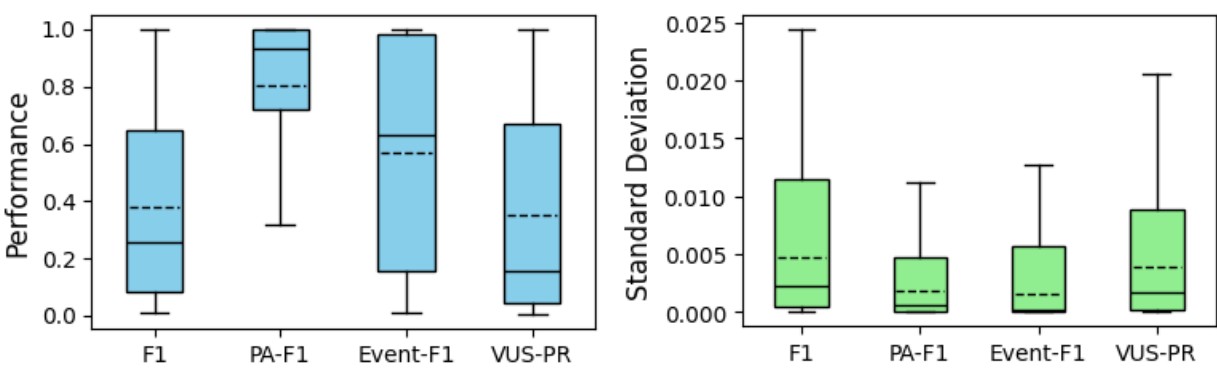

Figure 8: Distribution of standard deviation of four evaluation measures for time-series anomaly detection task on TSB-AD benchmark (comprising 180 time series). Results come from five random seeds. The mean is marked by a dashed line and the median by a solid line.

## B.2 Full Time-Series Analysis Results

We provide complete evaluation results for time-series classification in Table 9, anomaly detection in Table 10, 11, forecasting in Table 12, few-shot learning in Table 13 and zero-shot learning in Table 14.

Table 9: Performance overview on the classification task on UEA datasets (Bagnall et al., 2018). *. in the Transformers indicates the name of *former. The best performance is highlighted in **bold**, and the second-best is underlined.

| Category | Model | EC | FD | HW | HB | JV | PSF | SRSCP1 | SRSCP2 | SAD | UWGL | Average |
|---|---|---|---|---|---|---|---|---|---|---|---|---|
| Classical | XGBoost (2016) | 43.7 | 63.3 | 15.8 | 73.2 | 86.5 | **98.3** | 84.6 | 48.9 | 69.6 | 75.9 | 66.0 |
| | Rocket (2020) | **45.2** | 64.7 | **58.8** | 75.6 | 96.2 | 75.1 | 90.8 | 53.3 | 71.2 | **94.4** | 72.5 |
| RNN | LSTNet (2018) | 39.9 | 65.7 | 25.8 | 77.1 | 98.1 | 86.7 | 84.0 | 52.8 | **100.0** | 87.8 | 71.8 |
| | LSSL (2021) | 31.1 | 66.7 | 24.6 | 72.7 | 98.4 | 86.1 | 90.8 | 52.2 | 100.0 | 85.9 | 70.9 |
| CNN | TCN (2019) | 28.9 | 52.8 | 53.3 | 75.6 | 98.9 | 68.8 | 84.6 | 55.6 | 95.6 | 88.4 | 70.3 |
| | TimesNet (2023) | 35.7 | 68.6 | 32.1 | 78.0 | 98.4 | 89.6 | 91.8 | 57.2 | 99.0 | 85.3 | 73.6 |
| Transformers | Trans. (2017) | 32.7 | 67.3 | 32.0 | 76.1 | 98.7 | 82.1 | 92.2 | 53.9 | 98.4 | 85.6 | 71.9 |
| | Re. (2020) | 31.9 | 68.6 | 27.4 | 77.1 | 97.8 | 82.7 | 90.4 | 56.7 | 97.0 | 85.6 | 71.5 |
| | In. (2021) | 31.6 | 67.0 | 32.8 | **80.5** | 98.9 | 81.5 | 90.1 | 53.3 | **100.0** | 85.6 | 72.1 |
| | Pyra. (2021) | 30.8 | 65.7 | 29.4 | 75.6 | 98.4 | 83.2 | 88.1 | 53.3 | 99.6 | 83.4 | 70.8 |
| | Auto. (2021) | 31.6 | 68.4 | 36.7 | 74.6 | 96.2 | 82.7 | 84.0 | 50.6 | **100.0** | 85.9 | 71.1 |
| | Station. (2022) | 32.7 | 68.0 | 31.6 | 73.7 | 99.2 | 87.3 | 89.4 | 57.2 | **100.0** | 87.5 | 72.7 |
| | FED. (2022) | 31.2 | 66.0 | 28.0 | 73.7 | 98.4 | 80.9 | 88.7 | 54.4 | **100.0** | 85.3 | 70.7 |
| | ETS. (2022) | 28.1 | 66.3 | 32.5 | 71.2 | 95.9 | 86.0 | 89.6 | 55.0 | **100.0** | 85.0 | 71.0 |
| | Flow. (2022) | 33.8 | 67.6 | 33.8 | 77.6 | 98.9 | 83.8 | 92.5 | 56.1 | 98.8 | 86.6 | 73.0 |
| MLP | DLinear (2023) | 32.6 | 68.0 | 27.0 | 75.1 | 96.2 | 75.1 | 87.3 | 50.5 | 81.4 | 82.1 | 67.5 |
| | LightTS (2022) | 29.7 | 67.5 | 26.1 | 75.1 | 96.2 | 88.4 | 89.8 | 51.1 | **100.0** | 80.3 | 70.4 |
| FM | MOMENT (2024) | 35.7 | 63.3 | 30.8 | 72.2 | 71.6 | 89.6 | 84.0 | 47.8 | 98.1 | 90.9 | 68.4 |
| | UniTS (2024) | 37.6 | **70.5** | 29.7 | 80.0 | 97.8 | 93.1 | **93.9** | **61.1** | 98.9 | 87.8 | 75.0 |
| | OFA (2023) | 33.1 | 69.2 | 30.9 | 78.0 | 82.4 | 87.9 | 93.5 | 60.1 | 99.3 | 86.9 | 72.2 |
| | **MLLM4TS(Ours)** | 38.8 | 68.5 | 40.0 | 80.0 | **99.7** | 94.2 | 93.2 | 60.6 | 99.6 | 92.8 | **76.7** |

Table 10: Performance overview on the anomaly detection task on TSB-AD-M benchmark (Liu & Paparrizos, 2024). Performance is evaluated in VUS-PR (Paparrizos et al., 2022; Boniol et al., 2025). The best performance is highlighted in **bold**, and the second-best is underlined.

| Dataset | Statistical | | | | | NN | | | | | FM | |
|---|---|---|---|---|---|---|---|---|---|---|---|---|
| | PCA (2017) | KMeansAD (2001) | CBLOF (2003) | MCD (1999) | OCSVM (1999) | CNN (2018) | OmniAnomaly (2019) | LSTMAD (2015) | USAD (2020) | AutoEncoder (2014) | OFA (2023) | MLLM4TS (Ours) |
| GHL | 0.012 | 0.030 | 0.019 | 0.014 | 0.036 | 0.062 | **0.065** | 0.062 | **0.065** | 0.047 | 0.007 | 0.007 |
| Genesis | 0.019 | **0.891** | 0.024 | 0.059 | 0.076 | 0.100 | 0.003 | 0.037 | 0.003 | 0.007 | 0.013 | 0.017 |
| SWaT | 0.449 | 0.159 | 0.292 | 0.538 | 0.444 | 0.150 | 0.150 | 0.156 | 0.150 | **0.575** | 0.139 | 0.183 |
| SMAP | 0.093 | **0.380** | 0.137 | 0.104 | 0.116 | 0.193 | 0.124 | 0.163 | 0.108 | 0.129 | 0.208 | 0.348 |
| MSL | 0.149 | **0.435** | 0.215 | 0.229 | 0.216 | 0.217 | 0.217 | 0.217 | 0.226 | 0.219 | 0.212 | 0.237 |
| GECCO | 0.202 | 0.055 | 0.034 | 0.033 | 0.038 | 0.303 | 0.021 | 0.019 | 0.021 | 0.049 | 0.000 | **0.712** |
| CATSv2 | 0.118 | 0.117 | 0.059 | 0.132 | 0.080 | 0.080 | 0.041 | 0.041 | 0.041 | 0.063 | 0.049 | 0.105 |
| Daphnet | 0.130 | 0.297 | 0.096 | 0.135 | 0.064 | 0.203 | 0.340 | 0.311 | 0.340 | 0.129 | **0.378** | 0.338 |
| OPP | **0.299** | 0.063 | 0.143 | 0.167 | 0.121 | 0.177 | 0.177 | 0.165 | 0.177 | 0.144 | 0.072 | 0.091 |
| Exathlon | **0.949** | 0.372 | 0.857 | 0.796 | 0.830 | 0.684 | 0.839 | 0.816 | 0.839 | 0.909 | 0.865 | 0.879 |
| SMD | 0.364 | 0.358 | 0.223 | 0.260 | 0.285 | 0.174 | 0.325 | 0.325 | 0.160 | 0.301 | 0.398 | **0.455** |
| PSM | 0.163 | 0.208 | 0.194 | 0.255 | 0.191 | 0.236 | 0.160 | 0.236 | 0.194 | **0.280** | 0.158 | 0.145 |
| CreditCard | 0.103 | 0.020 | 0.032 | 0.060 | 0.024 | 0.022 | 0.021 | 0.022 | 0.021 | 0.028 | **0.156** | 0.143 |
| MITDB | 0.065 | 0.063 | 0.039 | 0.037 | 0.038 | 0.115 | 0.092 | 0.092 | **0.118** | 0.038 | 0.032 | 0.112 |
| SVDB | 0.112 | 0.203 | 0.068 | 0.067 | 0.065 | **0.352** | 0.155 | 0.155 | 0.322 | 0.065 | 0.096 | 0.117 |
| LTDB | 0.244 | 0.414 | 0.202 | 0.214 | 0.198 | 0.303 | **0.444** | 0.303 | 0.411 | 0.206 | 0.134 | 0.287 |
| TAO | **1.000** | 0.862 | **1.000** | **1.000** | 0.810 | 0.991 | 0.813 | 0.991 | 0.813 | 0.997 | 0.909 | **1.000** |
| **TSB-AD-M** | 0.310 | 0.295 | 0.273 | 0.271 | 0.265 | 0.312 | 0.312 | 0.307 | 0.304 | 0.295 | 0.296 | **0.349** |

Table 11: Performance overview on the anomaly detection task on four common datasets. Performance is evaluated in point-adjusted F-score. The best performance is highlighted in **bold**, and the second-best is underlined.

| Dataset | MLLM4TS (Ours) | OFA (2023) | TimesNet (2023) | PatchTS. (2022) | ETS. (2022) | FED. (2022) | LightTS (2022) | DLinear (2023) | Station. (2022) | Auto. (2021) | Pyra. (2021) | In. (2021) | Re. (2020) | Trans. (2017) |
|---|---|---|---|---|---|---|---|---|---|---|---|---|---|---|
| SMD | **87.4** | 86.9 | 84.6 | 84.6 | 83.1 | 85.1 | 82.5 | 77.1 | 84.7 | 85.1 | 83.0 | 81.7 | 75.3 | 79.6 |
| MSL | **90.8** | 81.8 | 81.8 | 78.7 | 85.0 | 78.6 | 79.0 | 84.9 | 77.5 | 79.1 | 84.9 | 84.1 | 84.4 | 78.7 |
| SMAP | **78.4** | 68.8 | 69.4 | 68.8 | 69.5 | 70.8 | 69.2 | 69.3 | 71.1 | 71.1 | 71.1 | 69.9 | 70.4 | 69.7 |
| SWaT | **95.5** | 95.1 | 93.0 | 85.7 | 84.9 | 93.2 | 93.3 | 87.5 | 79.9 | 92.7 | 91.8 | 81.4 | 82.8 | 80.4 |
| PSM | **97.6** | 97.1 | 97.3 | 96.1 | 91.8 | 97.2 | 97.2 | 93.6 | 97.3 | 93.3 | 82.1 | 77.1 | 73.6 | 76.1 |
| **Average** | **89.9** | 85.9 | 85.2 | 82.8 | 82.9 | 85.0 | 84.2 | 82.5 | 82.1 | 84.3 | 82.6 | 78.8 | 77.3 | 76.9 |

Table 12: Performance overview on the forecasting task. The best performance is highlighted in **bold**, and the second-best is underlined.

| Method | Metric | MLLM4TS (Ours) MSE | MAE | OFA (2023) MSE | MAE | TimeVLM (2025) MSE | MAE | VisionTS (2025) MSE | MAE | AutoTimes (2024e) MSE | MAE | TimeLLM (2024) MSE | MAE | UniTime (2024d) MSE | MAE | iTrans. (2023) MSE | MAE | DLinear (2023) MSE | MAE | PatchTST (2022) MSE | MAE | TimesNet (2023) MSE | MAE |
|---|---|---|---|---|---|---|---|---|---|---|---|---|---|---|---|---|---|---|---|---|---|---|---|
| Weather 96 | | 0.149 | 0.198 | 0.154 | 0.205 | 0.149 | 0.201 | **0.144** | **0.196** | 0.158 | 0.208 | 0.149 | 0.200 | 0.180 | 0.223 | 0.163 | 0.211 | 0.152 | 0.237 | 0.149 | 0.198 | 0.172 | 0.220 |
| Weather 192 | | **0.193** | 0.245 | 0.196 | 0.245 | 0.194 | 0.243 | 0.196 | 0.243 | 0.207 | 0.254 | 0.195 | 0.243 | 0.226 | 0.261 | 0.205 | 0.250 | 0.220 | 0.282 | 0.194 | **0.241** | 0.219 | 0.261 |
| Weather 336 | | **0.243** | **0.282** | 0.254 | 0.290 | 0.245 | **0.282** | 0.265 | 0.295 | 0.262 | 0.298 | 0.245 | **0.282** | 0.280 | 0.300 | 0.254 | 0.289 | 0.265 | 0.319 | 0.245 | **0.282** | 0.280 | 0.306 |
| Weather 720 | | 0.315 | 0.337 | 0.321 | 0.337 | 0.318 | 0.336 | 0.337 | 0.342 | 0.342 | 0.353 | 0.318 | 0.338 | 0.355 | 0.348 | 0.329 | 0.340 | 0.323 | 0.362 | **0.314** | **0.334** | 0.365 | 0.359 |
| Weather Avg | | **0.225** | 0.266 | 0.231 | 0.269 | 0.227 | 0.266 | 0.236 | 0.269 | 0.242 | 0.278 | 0.227 | 0.266 | 0.260 | 0.283 | 0.238 | 0.272 | 0.240 | 0.300 | 0.226 | **0.264** | 0.259 | 0.287 |
| Solar 96 | | **0.167** | 0.231 | 0.196 | 0.261 | 0.169 | 0.233 | 0.213 | 0.241 | 0.171 | **0.221** | 0.224 | 0.289 | 0.223 | 0.274 | 0.187 | 0.255 | 0.191 | 0.256 | 0.168 | 0.237 | 0.178 | 0.256 |
| Solar 192 | | 0.185 | 0.245 | 0.224 | 0.292 | **0.179** | 0.242 | 0.233 | 0.262 | 0.190 | **0.236** | 0.244 | 0.289 | 0.251 | 0.290 | 0.200 | 0.270 | 0.211 | 0.273 | 0.187 | 0.263 | 0.200 | 0.268 |
| Solar 336 | | **0.192** | 0.251 | 0.240 | 0.308 | 0.195 | 0.254 | 0.236 | 0.270 | 0.203 | **0.248** | 0.225 | 0.291 | 0.270 | 0.301 | 0.209 | 0.276 | 0.228 | 0.287 | 0.196 | 0.260 | 0.212 | 0.274 |
| Solar 720 | | 0.209 | **0.257** | 0.256 | 0.321 | 0.204 | 0.265 | 0.241 | 0.289 | 0.222 | 0.262 | 0.243 | 0.301 | 0.271 | 0.298 | 0.213 | 0.276 | 0.236 | 0.295 | **0.205** | 0.269 | 0.211 | 0.273 |
| Solar Avg | | 0.188 | 0.246 | 0.229 | 0.296 | **0.187** | 0.249 | 0.231 | 0.266 | 0.197 | **0.242** | 0.234 | 0.293 | 0.254 | 0.291 | 0.202 | 0.269 | 0.217 | 0.278 | 0.189 | 0.257 | 0.200 | 0.268 |
| ETTh1 96 | | 0.366 | 0.400 | 0.377 | 0.404 | 0.365 | 0.395 | **0.343** | **0.376** | 0.360 | 0.397 | 0.380 | 0.412 | 0.386 | 0.409 | 0.386 | 0.405 | 0.375 | 0.399 | 0.370 | 0.399 | 0.384 | 0.402 |
| ETTh1 192 | | 0.404 | 0.420 | 0.413 | 0.424 | 0.401 | 0.418 | **0.379** | **0.405** | 0.391 | 0.419 | 0.405 | 0.422 | 0.428 | 0.436 | 0.422 | 0.439 | 0.405 | 0.416 | 0.413 | 0.421 | 0.557 | 0.436 |
| ETTh1 336 | | 0.425 | 0.434 | 0.436 | 0.444 | 0.440 | 0.456 | 0.412 | **0.423** | **0.408** | 0.432 | 0.422 | 0.433 | 0.464 | 0.456 | 0.444 | 0.457 | 0.439 | 0.443 | 0.422 | 0.436 | 0.491 | 0.469 |
| ETTh1 720 | | 0.436 | 0.467 | 0.477 | 0.481 | 0.536 | 0.538 | 0.458 | 0.455 | **0.429** | **0.452** | 0.430 | 0.459 | 0.473 | 0.479 | 0.500 | 0.498 | 0.472 | 0.490 | 0.447 | 0.466 | 0.521 | 0.500 |
| ETTh1 Avg | | 0.408 | 0.430 | 0.426 | 0.438 | 0.436 | 0.452 | 0.398 | **0.415** | **0.397** | 0.425 | 0.409 | 0.432 | 0.438 | 0.445 | 0.438 | 0.450 | 0.423 | 0.437 | 0.413 | 0.431 | 0.458 | 0.450 |
| ECL 96 | | 0.134 | 0.232 | 0.137 | 0.236 | 0.139 | 0.242 | **0.126** | 0.223 | 0.140 | 0.236 | 0.137 | 0.244 | 0.171 | 0.266 | 0.132 | 0.227 | 0.153 | 0.237 | 0.129 | **0.222** | 0.168 | 0.272 |
| ECL 192 | | 0.153 | 0.251 | 0.154 | 0.251 | 0.154 | 0.255 | **0.144** | 0.241 | 0.159 | 0.253 | 0.162 | 0.271 | 0.178 | 0.274 | 0.152 | 0.249 | 0.152 | 0.249 | 0.147 | **0.240** | 0.184 | 0.289 |
| ECL 336 | | 0.169 | 0.267 | 0.169 | 0.267 | 0.171 | 0.271 | **0.163** | 0.255 | 0.177 | 0.270 | 0.175 | 0.279 | 0.194 | 0.289 | 0.167 | 0.262 | 0.169 | 0.267 | **0.163** | 0.259 | 0.198 | 0.300 |
| ECL 720 | | 0.204 | 0.295 | 0.207 | 0.300 | 0.208 | 0.301 | 0.195 | 0.286 | 0.216 | 0.303 | 0.207 | 0.306 | 0.232 | 0.319 | **0.192** | **0.285** | 0.233 | 0.344 | 0.197 | 0.290 | 0.220 | 0.320 |
| ECL Avg | | 0.165 | 0.261 | 0.167 | 0.264 | 0.168 | 0.267 | **0.157** | **0.251** | 0.173 | 0.266 | 0.170 | 0.275 | 0.194 | 0.287 | 0.161 | 0.256 | 0.177 | 0.274 | 0.159 | 0.253 | 0.192 | 0.295 |
| Traffic 96 | | 0.378 | 0.268 | 0.395 | 0.283 | 0.389 | 0.284 | 0.356 | **0.245** | 0.369 | 0.257 | 0.373 | 0.280 | 0.438 | 0.291 | **0.351** | 0.257 | 0.410 | 0.282 | 0.360 | 0.249 | 0.593 | 0.321 |
| Traffic 192 | | 0.396 | 0.281 | 0.410 | 0.290 | 0.399 | 0.285 | 0.385 | **0.251** | 0.394 | 0.268 | 0.390 | 0.288 | 0.446 | 0.293 | **0.364** | 0.265 | 0.423 | 0.287 | 0.379 | 0.256 | 0.617 | 0.336 |
| Traffic 336 | | 0.404 | 0.280 | 0.414 | 0.295 | 0.412 | 0.292 | 0.398 | **0.262** | 0.413 | 0.278 | 0.407 | 0.299 | 0.461 | 0.300 | **0.382** | 0.273 | 0.436 | 0.296 | 0.392 | 0.264 | 0.629 | 0.336 |
| Traffic 720 | | 0.446 | 0.304 | 0.445 | 0.311 | 0.481 | 0.335 | 0.439 | **0.284** | 0.449 | 0.299 | 0.438 | 0.310 | 0.494 | 0.318 | **0.420** | 0.292 | 0.466 | 0.315 | 0.432 | 0.286 | 0.640 | 0.350 |
| Traffic Avg | | 0.406 | 0.283 | 0.416 | 0.295 | 0.420 | 0.299 | 0.395 | **0.261** | 0.406 | 0.276 | 0.402 | 0.294 | 0.460 | 0.301 | **0.379** | 0.272 | 0.434 | 0.295 | 0.391 | 0.264 | 0.620 | 0.336 |

Table 13: Performance comparison between MLLM4TS and OFA (Zhou et al., 2023) on few-shot forecasting. 10% of the training data is used to train the model. We mark the better performance in **bold**.

| Method | Horizon | MLLM4TS MSE | MLLM4TS MAE | OFA MSE | OFA MAE |
|---|---|---|---|---|---|
| | Metric | | | | |
| Weather | 96 | 0.164 | 0.219 | **0.161** | **0.212** |
| | 192 | **0.202** | **0.247** | 0.207 | 0.253 |
| | 336 | **0.258** | **0.295** | 0.264 | 0.298 |
| | 720 | 0.322 | 0.338 | **0.321** | **0.335** |
| | Avg | **0.236** | **0.274** | 0.238 | 0.275 |
| ETTh1 | 96 | 0.494 | 0.489 | **0.464** | **0.472** |
| | 192 | **0.522** | 0.509 | 0.526 | **0.507** |
| | 336 | **0.534** | **0.511** | 0.747 | 0.601 |
| | 720 | **0.761** | 0.633 | 0.769 | **0.632** |
| | Avg | **0.578** | **0.535** | 0.626 | 0.553 |

Table 14: Performance overview on zero-shot forecasting. For MLLM4TS and OFA, the model is trained on ETTh2 dataset and then tested on ETTh1 dataset. For time series foundation models pretrained from time series corpus, the models are directly applied on ETTh1 dataset. The best performance is highlighted in **bold**, and the second-best is underlined.

| Dataset | Horizon | MLLM4TS MSE | MAE | OFA (2023) MSE | MAE | Chronos (2024) MSE | MAE | Moirai (2024) MSE | MAE | MOMENT (2024) MSE | MAE | LLMTime (2023) MSE | MAE |
|---|---|---|---|---|---|---|---|---|---|---|---|---|---|
| ETTh2→ETTh1 | 96 | 0.503 | 0.488 | 0.459 | 0.458 | 0.441 | 0.390 | **0.381** | **0.388** | 0.688 | 0.557 | 1.130 | 0.777 |
| | 192 | 0.454 | 0.459 | 0.496 | 0.481 | 0.502 | 0.524 | **0.434** | **0.415** | 0.688 | 0.560 | 1.242 | 0.820 |
| | 336 | 0.518 | 0.496 | 0.537 | 0.517 | 0.576 | 0.467 | **0.485** | **0.445** | 0.675 | 0.563 | 1.328 | 0.864 |
| | 720 | **0.521** | 0.517 | 0.604 | 0.556 | 0.835 | 0.583 | 0.611 | **0.510** | 0.683 | 0.585 | 4.145 | 1.461 |
| | Avg | 0.499 | 0.490 | 0.524 | 0.503 | 0.588 | 0.466 | **0.480** | **0.430** | 0.683 | 0.560 | 1.961 | 0.981 |

## B.3 Ablation Study Results

We present ablation studies across multiple aspects of different time-series analysis tasks, including classification (Table 15), anomaly detection (Table 16), and forecasting (Table 17). In addition, Figure 10 illustrates the effect of different language model backbones.

To better understand how the language model processes multimodal input, we visualize the attention maps of the language model backbone in Figure 9. The input consists of a concatenation of time-series tokens $(TS_1, \ldots, TS_N)$ and visual tokens $(V_1, \ldots, V_M)$. In the early transformer layers, attention is primarily concentrated on the time-series tokens, with limited attention directed toward the visual tokens. As the depth of the model increases, attention becomes more evenly distributed across both modalities, indicating that cross-modal interactions become more prominent in the deeper layers of the language model.

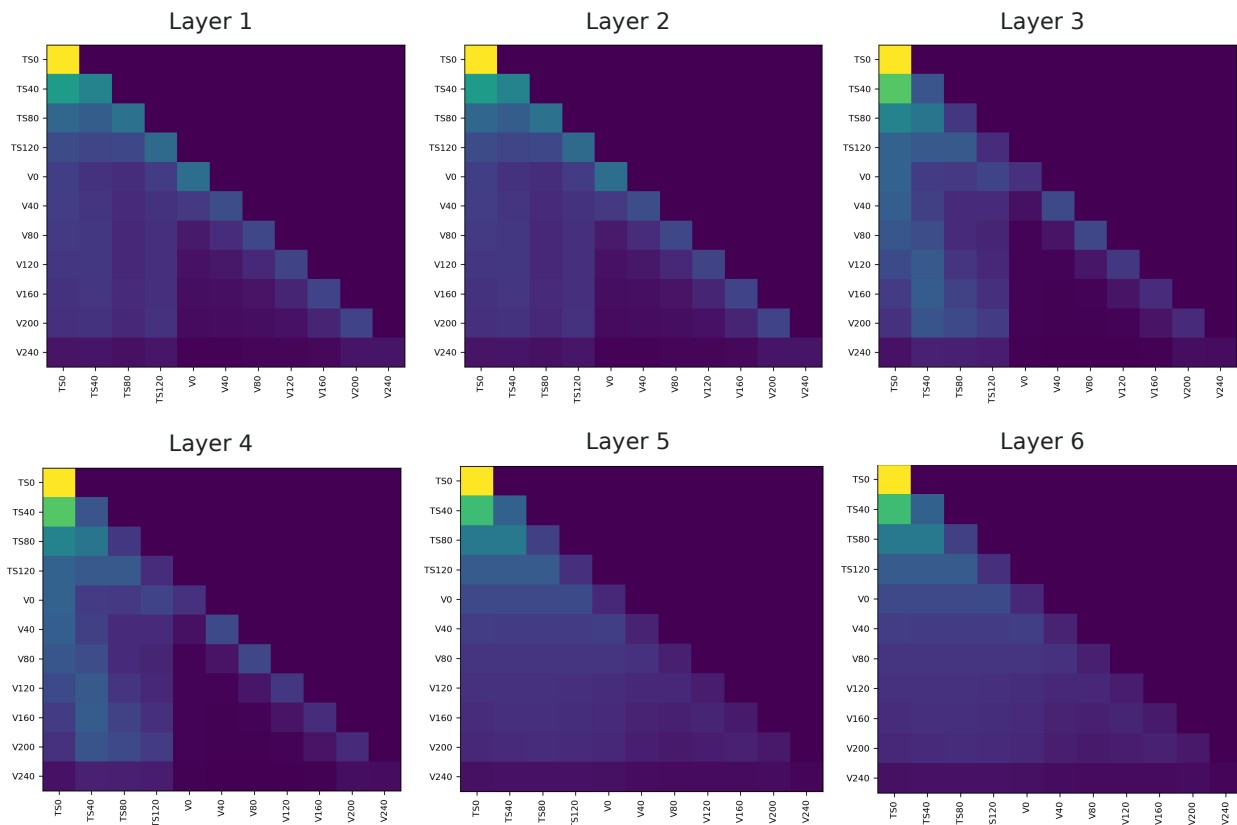

Figure 9: Layer-wise self-attention maps of the language model in MLLM4TS (Late) for the classification task. Each heatmap displays the average attention weights over the entire test set. The input tokens are concatenated time-series tokens $(TS_1, \ldots, TS_N)$ and visual tokens $(V_1, \ldots, V_M)$.

The consistent performance improvements over the time-series-only counterpart underscore the effectiveness of incorporating a vision modality into time-series analysis. Ablation results across different model variants further validate the robustness of the proposed framework. An alternative to LLM-based direct forecasting is the autoregressive approach, where future values are generated sequentially based on previously predicted outputs, as suggested by recent work (Liu et al., 2024e). As shown in Table 17 ('AutoReg'), this method performs well for short prediction horizons but suffers from error accumulation as the forecasting window lengthens, leading to degraded performance. Moreover, increasing the language model size from GPT-2 (Radford et al., 2019) (124M parameters) to larger models such as Qwen3 (Yang et al., 2024) (1.7B parameters) does not yield further gains in multimodal time-series tasks. These findings suggest that smaller

models provide sufficient language modeling capacity, while larger models may be more prone to overfitting noise in time-series data, potentially hindering generalization.

Table 15: Ablation study on time-series classification task. 'LLM2Attn' replaces the language model with one single attention layer. 'Layout' replaces horizontal layout with grid layout. 'VisualEnc' replaces CLIP with ResNet. 'Fusion' replaces early fusion with late fusion.

| Dataset | TS-Only | | Plot-TS | | | | |
|---|---|---|---|---|---|---|---|
| | OFA | LLM2Attn | MLLM4TS | Layout | VisualEnc | Fusion | LLM2Attn |
| EC | 33.1 | 33.1 | 38.8 | 41.1 | 34.1 | 35.0 | 33.8 |
| FD | 69.2 | 66.1 | 68.5 | 58.7 | 68.6 | 58.0 | 60.2 |
| HW | 30.9 | 32.9 | 40.0 | 43.1 | 34.2 | 33.9 | 32.6 |
| HB | 78.0 | 75.6 | 80.0 | 80.5 | 78.1 | 80.0 | 77.6 |
| JV | 82.4 | 93.8 | 99.7 | 99.2 | 98.7 | 99.2 | 98.4 |
| PSF | 87.9 | 89.0 | 94.2 | 89.0 | 85.6 | 86.7 | 83.8 |
| SRSCP1 | 93.5 | 93.2 | 93.2 | 91.1 | 90.1 | 89.8 | 91.8 |
| SRSCP2 | 60.1 | 57.2 | 60.6 | 62.2 | 57.2 | 64.4 | 56.7 |
| SAD | 99.3 | 73.8 | 99.6 | 99.1 | 87.7 | 96.3 | 87.3 |
| UWGL | 86.9 | 87.2 | 92.8 | 87.9 | 91.9 | 91.3 | 91.6 |
| **Average** | 72.2 | 70.2 | 76.7 | 75.2 | 72.6 | 73.5 | 71.4 |

Table 16: Ablation study on time-series anomaly detection task. 'LLM2Attn' replaces the language model with one single attention layer. 'Layout' replaces horizontal layout with grid layout. 'VisualEnc' replaces CLIP with ResNet. 'Fusion' replaces early fusion with late fusion.

| Domain | TS-Only | | Plot-TS | | | | |
|---|---|---|---|---|---|---|---|
| | OFA | LLM2Attn | MLLM4TS | Layout | VisualEnc | Fusion | LLM2Attn |
| Environment | 0.909 | 1.000 | 1.000 | 1.000 | 1.000 | 1.000 | 0.886 |
| Facility | 0.647 | 0.599 | 0.679 | 0.678 | 0.692 | 0.679 | 0.692 |
| Finance | 0.156 | 0.154 | 0.143 | 0.145 | 0.145 | 0.149 | 0.222 |
| HumanActivity | 0.110 | 0.102 | 0.122 | 0.120 | 0.122 | 0.120 | 0.092 |
| Medical | 0.083 | 0.085 | 0.131 | 0.134 | 0.139 | 0.133 | 0.163 |
| Sensor | 0.125 | 0.117 | 0.194 | 0.180 | 0.181 | 0.179 | 0.164 |
| **Average** | 0.296 | 0.286 | 0.349 | 0.344 | 0.348 | 0.343 | 0.340 |

Table 17: Ablation study on time-series forecasting task. 'AutoReg' trains the model and generates forecasting results in an autoregressive manner as described in AutoTimes (Liu et al., 2024e). 'VisualEnc' replaces CLIP with ResNet. 'Fusion' replaces early fusion with late fusion. 'LLM2Attn' replaces the language model with one single attention layer.

| Dataset | ETTh1 | | | | | | | | Weather | | | | | | | |
|---|---|---|---|---|---|---|---|---|---|---|---|---|---|---|---|---|
| Type | **MLLM4TS** | | AutoReg | | VisualEnc | | LLM2Attn | | **MLLM4TS** | | AutoReg | | VisualEnc | | LLM2Attn | |
| Metric | MSE | MAE | MSE | MAE | MSE | MAE | MSE | MAE | MSE | MAE | MSE | MAE | MSE | MAE | MSE | MAE |
| Pred-96 | 0.366 | 0.4 | **0.361** | **0.394** | 0.397 | 0.423 | 0.479 | 0.483 | **0.149** | **0.198** | **0.149** | 0.201 | 0.165 | 0.217 | 0.187 | 0.243 |
| Pred-192 | 0.404 | 0.42 | **0.397** | **0.417** | 0.436 | 0.593 | 0.495 | 0.509 | **0.193** | **0.245** | 0.202 | 0.248 | 0.227 | 0.271 | 0.225 | 0.273 |
| Pred-336 | 0.425 | 0.434 | **0.42** | **0.433** | 0.475 | 0.477 | 0.506 | 0.522 | **0.243** | **0.282** | 0.263 | 0.291 | 0.269 | 0.307 | 0.267 | 0.303 |
| Pred-720 | **0.436** | 0.467 | 0.446 | **0.46** | 0.515 | 0.514 | 0.540 | 0.561 | **0.315** | **0.337** | 0.336 | 0.343 | 0.331 | 0.352 | 0.328 | 0.347 |

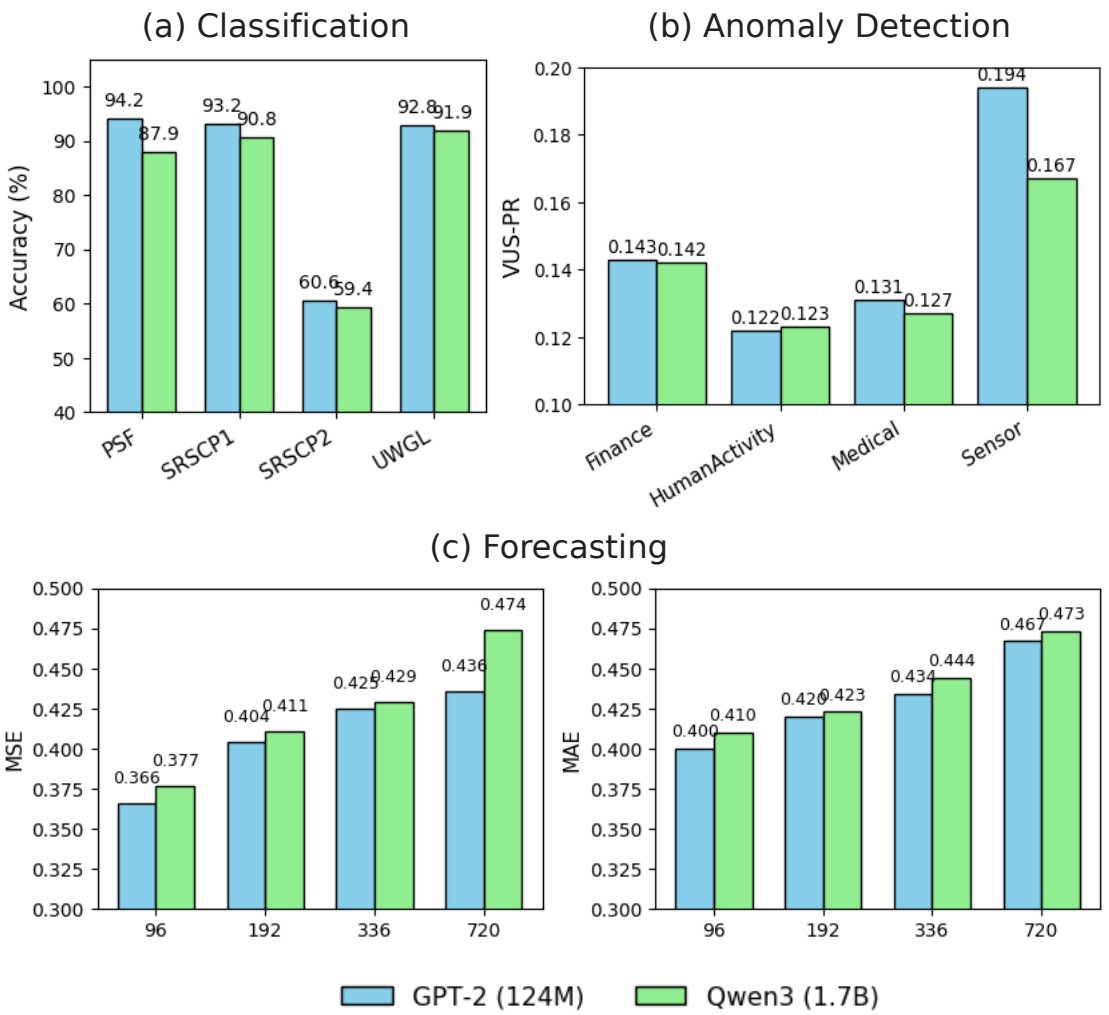

Figure 10: Comparison of task performance using different language model backbones: GPT-2 (Radford et al., 2019) (124M parameters) and Qwen3 (Yang et al., 2024) (1.7B parameters). Panels (a) classification and (b) anomaly detection show metrics for which higher values indicate better performance, while panel (c) forecasting presents forecasting errors, where lower values denote better results.

## C    Show Case

We provide an illustration of horizontal and grid layout in Figure 11, and example plots of datasets used in this work in Figure 12.

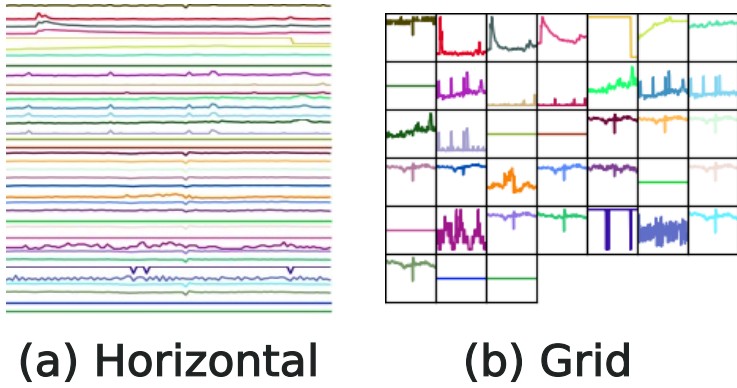

Figure 11: Example plots of (a) horizontal and (b) grid layout.

## D    Broader Impact

**Impact on Real-world Applications.** MLLM4TS offers a unified and effective solution for a wide range of time series analysis tasks, including but not limited to classification, anomaly detection, and forecasting. These capabilities support practical applications in domains such as electrocardiogram monitoring, human activity recognition, financial modeling, facility management, environmental sensing, and industrial process control. Its strong performance in few-shot and zero-shot scenarios, together with its robustness to hyperparameter variation (such as patch size), highlights its reliability in data-sparse environments. These characteristics make MLLM4TS a strong candidate for deployment in decision support systems across healthcare, manufacturing, finance, and climate-related services.

**Impact on Future Research.** This work is among the first to incorporate line plots, an intuitive and widely used method for visualizing time series, into the context of multi-modal time series analysis. While the proposed framework focuses on line plot representations, its architecture can be extended to incorporate aligned image and video data. Furthermore, our investigation into the roles of visual representations and language model backbones provides valuable insights for the development of explainable and agentic time series analysis, paving the way for safer and more transparent deployment in high-stakes domains.

## E    Limitation and Future Work

One limitation of MLLM4TS is the additional computational overhead introduced by the vision branch, which increases runtime during the processing of visual embeddings. Future research may explore the design of more lightweight visual frontends for rendering visual representations, with the goal of improving runtime efficiency. Another promising direction is extending MLLM4TS to handle irregularly sampled time series. Given the demonstrated benefits of visual representations, converting such data into image form may offer a more natural solution, opening new avenues for multimodal time-series analysis in this context.

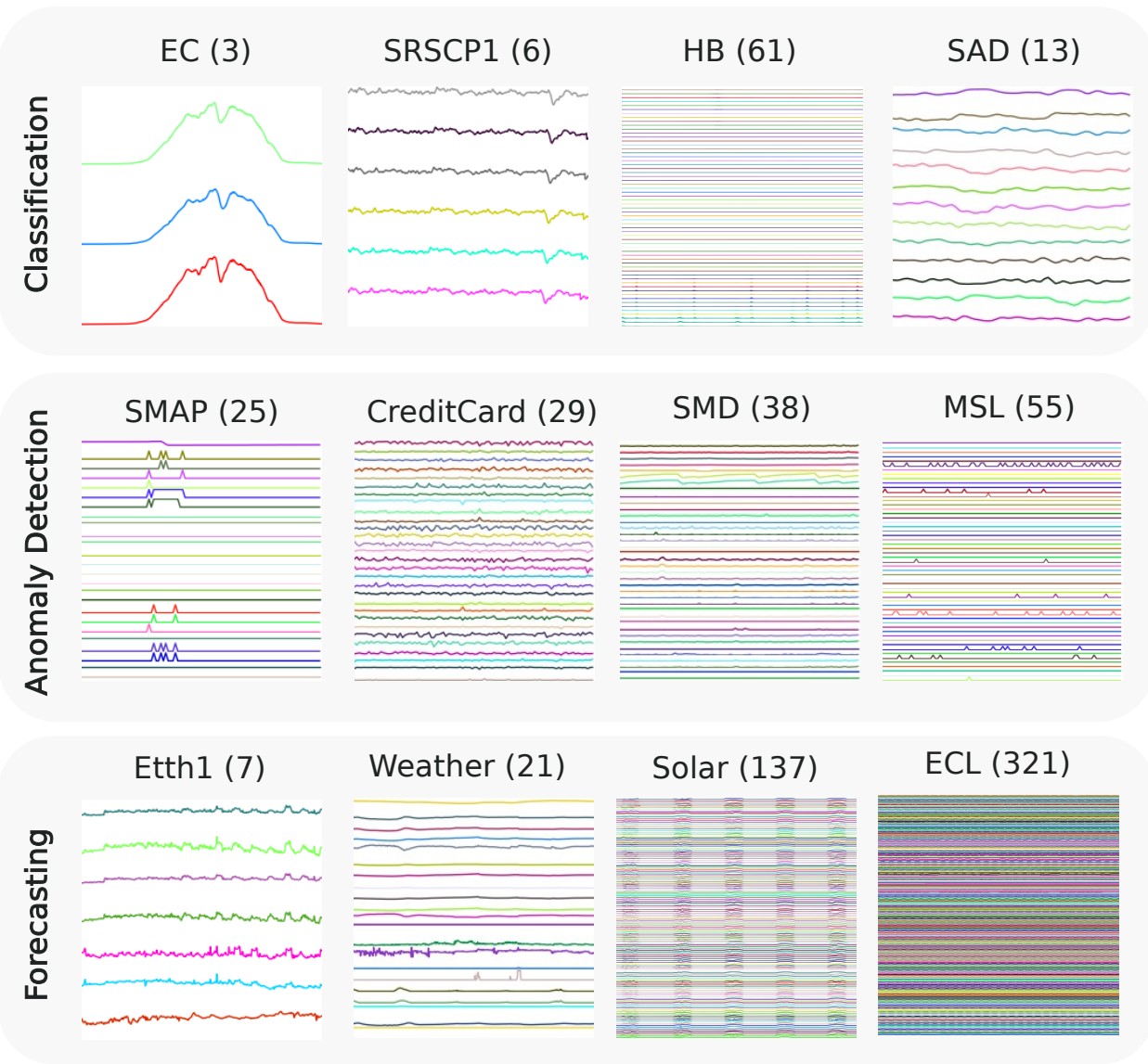

Figure 12: Example plots of an instance from datasets used in this work (dataset name and channel count in parentheses).

