# OpenReview forum: "MLLM4TS: Leveraging Vision and Multimodal Language Models for General Time-Series Analysis"
_TMLR — Decision pending for TMLR_

### Review · Reviewer_725x · 2026-04-30

**Summary Of Contributions:**

Given the array of domains where time series analysis is crucial, and its heterogenous, multichannel nature, a wide variety of solutions try to address the problem based on various techniques. This paper, motivated by the intuition of visual pattern recognition on time series charts aims to fuse such visual information into a capable sequence modeling LLM backbone which operates over time series patches. By transforming multivariate time series into colored  images, to capture global and cross-channel patterns MLLM4TS hopes to incorporate visual information into the prediction task. Functionally the paper achieves this by fusing Visual embeddings on the images via a pretrained image encoder and time-series embeddings framework to model temporal dynamics given the contextual information.
MLLM4TS tries to demonstrate its effectiveness via benchmarking over time-series classification, anomaly detection, and time series forecasting and tries to establish the necessity and effectiveness of their modality bridging and visuo-temporal alignment approach in this framework. A better reading of the model architecture is presented in Section 3 of the work.
The analysis of the work centers around 3 research questions (RQ's)
* RQ1. Does incorporating visual representations enhance the performance of general time-series
analysis tasks? The authors demonstrate some usefulness of embedding the visualized time series into the pipeline but they perform a very limited analysis on this front in their RQ2.
* RQ2. What types of visual representations (e.g., image layouts, visual encoders) are most effective
when integrated into the MLLM4TS framework?
* RQ3. Are language models actually useful for multi-modal time-series analysis ?
The authors conduct narrowly defined experiments to demonstrate the effectiveness of visual information encoding for time series data.

**Audience:**

Yes

**Audience Explanation:**

The paper proposes a clear way to incorporate additional information about a time series in the training pipeline. This would be useful to those who wish to incorporate exogenous like [4] or side information into the time series problem setting.
### References
[4] Time-Aware Prior Fitted Networks for Zero-Shot Forecasting with Exogenous Variables.

**Claims And Evidence:**

No

**Claims Explanation:**

In order to establish and test their methods effectiveness the paper focuses on very narrow variations/ablation tests.
The authors test line-plot images of the time series, with variants such as horizontal versus grid layout, visual encoder choice, and fusion strategy. This establishes that some additional information via plots can help but doesn't really allude to any possible hints for the mechanism behind it. A better version of this test will be to also compare againt FFTs or STFT and similar images of transform to see the changes in performance with variation along the axis of multi-modal inputs.
This will help us understand if any augmented representation to the time series is helpful, whether via extra compute available from finetuning or the capacity of the LLM to model more complicated relations or do line plots provide neccessarily meaningul information.
So far it can only be claimed that MLLM4TS learns from two representations of the same time series, rather than from truly independent multimodal sources.

For few shot comparison the model doesn't compare with prior fitted networks in Time series domain especially suited for this task like TimePFN[1], RATSF[2]. The latter seems competitive to the model whereas the former had a slightly different experimental protocol so direct comparison with reported numbers in the paper is difficult.

Regarding ablations the authors don't clarify why they chose LLM2Attn over LLM2Trsf as an ablation test as done in [3].

### References
[1] TimePFN: Effective Multivariate Time Series Forecasting with Synthetic Data. AAAI-2025 \
[2] Retrieval Augmented Time Series Forecasting, ICML 2025 \
[3] Are Language Models Actually Useful for Time Series Forecasting?

**Requested Changes:**

Please address concerns raised in the discussion for "Are the claims made in the submission supported by accurate, convincing and clear evidence?" Right now the paper’s “multimodality” claim is narrow, writing around ablations can be improved to contextualize what was tried and the paper needs to incorporate some relevant baselines from the PFN area to justify the usefulness of multi-modality via large models.

---

> ### Author Response · Authors · 2026-06-22
>
> We thank the reviewer for the rigorous reading and constructive suggestions. We would like to clarify that the current manuscript does not rely only on narrow ablations to demonstrate effectiveness. MLLM4TS is evaluated across classification, anomaly detection, and forecasting, including zero-/few-shot settings, and compared with strong baselines, including OFA, the closest LLM-only counterpart. The manuscript also includes ablations on visual representation, layout, encoder, fusion, patch size, and LLM replacement. We nevertheless agree that additional evidence can better isolate the source of the gain. We therefore add new visual-input controls, TimePFN, and a clearer LLM2Attn rationale.
>
> ### 1. Meaningful Visual Structure vs. Extra Representation
>
> To test whether the gain comes from meaningful visual structure rather than extra parameters or compute, we add a controlled ablation with the same visual encoder, fusion module, and training budget.
>
> | Visual input | Avg. classification accuracy |
> |---|---:|
> | Noise image | 66.0 |
> | Heatmap | 72.1 |
> | None / TS-only | 72.2 |
> | Line plot, Grid | 75.2 |
> | **Line plot, Horizontal, ours** | **76.7** |
>
> The noise image uses the same visual branch but contains no time-series information, and performs much worse than TS-only. This shows that the gain is not due to extra compute or parameters. The heatmap preserves the same channels, timestamps, values, and alignment, but performs nearly the same as TS-only, showing that not every visual encoding is sufficient. In contrast, both line-plot variants outperform TS-only, with the horizontal layout performing best. This suggests that pretrained vision encoders benefit from legible waveform structures such as trends, edges, motifs, and aligned temporal patterns.
>
> We use heatmaps rather than FFT/STFT as the main control because heatmaps preserve the same multivariate content while changing only the visual encoding. FFT/STFT would require extra design choices for per-channel aggregation, tiling, or fusion, and may weaken cross-channel temporal alignment. We will add this ablation and frame the visual branch as a complementary visual inductive-bias view, rather than claiming that any image transformation is sufficient.
>
> ### 2. Missing PFN / Few-Shot Baselines
>
> We add TimePFN [1] on the same few-shot splits, alongside OFA [2], the LLM-only baseline without the visual branch.
>
> | Dataset | MLLM4TS | OFA | TimePFN |
> |---|---:|---:|---:|
> | Weather Avg. MSE / MAE | **0.236 / 0.274** | 0.238 / 0.275 | 0.266 / 0.285 |
> | ETTh1 Avg. MSE / MAE | **0.578 / 0.535** | 0.626 / 0.553 | 0.675 / 0.555 |
>
> TimePFN trails MLLM4TS on both datasets, suggesting that the few-shot gain is not simply recovered by a strong non-LLM prior-fitted network on the same 10% training slice. The comparison with OFA further isolates the visual modality: OFA uses an LLM backbone without the visual branch, while MLLM4TS improves over OFA, especially on ETTh1. We will add TimePFN to the few-shot forecasting section and discuss RATSF separately, noting that its retrieval-augmented protocol is not directly comparable to our 10% few-shot setting.
>
> ### 3. Why LLM2Attn Rather Than LLM2Trsf?
>
> LLM2Attn corresponds to PAttn, the single-attention-layer replacement recommended by Tan et al. [3] for testing whether LLMs are useful for time-series forecasting. This makes it a stringent and directly relevant baseline, while preserving protocol fidelity with prior work.
>
> In the TS-only setting, we reproduce Tan et al.’s observation that LLM2Attn can be strong for forecasting. However, in the multimodal setting, the full LLM outperforms LLM2Attn across classification, anomaly detection, and forecasting. This suggests that the LLM becomes more useful when cross-modal integration is required. We will also connect this result to the scale ablation and Figure 9: increasing model size from GPT-2 to Qwen3-1.7B does not yield consistent gains, while deeper layers attend to both numerical and visual tokens. Thus, the gain is not simply a capacity effect.
>
> ### Summary of Revisions
>
> We will revise the manuscript to:
> 1. add a controlled visual ablation comparing different visual inputs;
> 2. frame the visual branch as a complementary visual inductive-bias view;
> 3. add TimePFN baseline;
> 4. clarify that LLM2Attn follows Tan et al.’s PAttn replacement;
> 5. connect the LLM ablation with scale analysis and attention visualization.
>
> These additions directly address whether the gain comes from meaningful line-plot structure rather than arbitrary added representation or compute. We thank the reviewer again for helping strengthen the empirical support and presentation.
>
> [1] Taga et al. “TimePFN: Effective Multivariate Time Series Forecasting with Synthetic Data.” AAAI, 2025.
> [2] Zhou et al. “One Fits All: Power General Time Series Analysis by Pretrained LM.” NeurIPS, 2023.
> [3] Tan et al. “Are Language Models Actually Useful for Time Series Forecasting?” NeurIPS, 2024.

---

### Review · Reviewer_ckj8 · 2026-05-15

**Summary Of Contributions:**

This paper introduces MLLM4TS, a multimodal framework for general time-series representations and various downstream time-series tasks. MLLM4TS uses both numerical time-series tokens and visual plot embeddings encoded by a visual encoder, fusing them before a pretrained language model for downstream time-series tasks. Experiments on classification, anomaly detection, forecasting, and few-/zero-shot settings show that the vision modality improves over multiple time-series baseline.

**Additional Comments:**

NA

**Audience:**

Yes

**Audience Explanation:**

Yes. The paper addresses the key questions about multimodal foundation models for time-series analysis, which should interest researchers working on time series, LLMs, CV, and multimodal learning.

**Broader Impact Concerns:**

No major broader impact concerns. The work is primarily methodological and evaluated on benchmark time-series tasks.

**Claims And Evidence:**

Yes

**Claims Explanation:**

Yes. The main claims are supported by experiments across classification, anomaly detection, forecasting, and few-/zero-shot settings. These results show that MLLM4TS achieves consistent improvements. The ablation studies further support the proposed design.

**Requested Changes:**

1. The authors should clarify the references for the visualization component. Since color-coded line plots are a common visualization strategy rather than a new method, appropriate prior work should be cited for this design choice.

2. The authors motivate the method by stating that anomalies can appear as "visually salient regions", and an example is the EKG inspection. However, the paper does not provide saliency-related analysis to support this claim. The authors should add appropriate references with the same finding, such as [1], to support this motivation claim.

[1] Bidirectional Generative Pre-training for Improving Healthcare Time-series Representation Learning. MLHC.

3. The paper may include saliency analysis in Introduction or Experiment sections to substantiate this motivation, similar to [1].

4. In Input Module section, this paper mentions that each channel has one uniquely color line plot. How does it selec color? What does color mean? Different color may have different impacts for performance? It lacks more discussion about this colored line plot setup.

5. In Time Series Tokenizer section, the claims about "facilitate knowledge transfer" is confounding as it does not do knowledge transfer for pretrained LM or multimodal foundaton models. I think it means that this work can helps transfer across time series with different distributions or something else？

6. The claim about no need for manual patch-size tuning is overstated. I think it is just using the image patch size P from a VIT encoder. This paper mentions that the selection of an appropriate patch size is non-trivial, which is not addressed. The reduced need for patch-size selection comes from using a foundation model. It is not clear whether it is appropriate or not, as it is predetermined by foundation model.

7. In Figure 6 , while color coding improves performance over the non-color-coded variant on most datasets, it is worse on EC and tied on SRSCP2; moreover, TS-only outperforms MLLM4TS on FD and SRSCP1. The authors should explain these dataset-specific differences with more discussions.

8. The authors should justify the choice of reducing high-dimensional time series to 50 plotted channels.

---

> ### Author Response · Authors · 2026-06-22
>
> We thank the reviewer for the supportive assessment and constructive suggestions. The requested changes mainly concern clarifying design choices, adding citations, and expanding under-explained components. We address each point below and will revise the manuscript accordingly.
>
> ### 1. References for Color-Coded Line Plots
>
> We agree that color-coded line plots are a standard visualization strategy and did not intend to claim them as a novel contribution. We will clarify this and add citations to prior work on visual time-series representations, building on Section 2. Our contribution is not the plot format itself, but integrating visual time-series embeddings with LLMs across classification, anomaly detection, and forecasting. We use color-coded plots because they are simple, interpretable, and broadly applicable.
>
> ### 2–3. Saliency Motivation for Visual Representation
>
> We will cite the suggested work [1], along with related studies, and soften the claim that anomalies appear as visually salient regions, since this depends on dataset characteristics. Figure 9 already shows that the LLM attends to both numerical and visual tokens, suggesting cross-modal usage. To provide more direct evidence, we will add pixel-level saliency maps.
>
> ### 4. Color Selection and Meaning
>
> We will clarify the color assignment rule in Section 3. Algorithm 1 in Appendix A.2 specifies `colors ← colormap(C)`, but this was not sufficiently emphasized. Each channel is assigned a color by channel index. Color encodes **channel identity**, not magnitude or semantic value, and improves visual separability among overlapping curves.
>
> The ablation in Figure 6 compares color-coded and non-color-coded variants. We will clarify that the gain mainly comes from making channels visually distinguishable, not from semantic meaning of specific colors. The exact palette is less important than sufficient contrast.
>
> ### 5. “Knowledge Transfer” in the Time-Series Tokenizer
>
> We apologize for the unclear wording. We did not mean that RevIN transfers knowledge into the pretrained LM or multimodal foundation model. Instead, RevIN helps handle distribution shifts across time series, such as differences in scale, offset, and temporal statistics. We will replace “facilitate knowledge transfer” with a precise explanation: RevIN normalizes and denormalizes each series to improve robustness across heterogeneous distributions.
>
> ### 6. Patch-Size Claim
>
> The original wording was meant to describe improved robustness to temporal patch-size choices, not to suggest that patch-size tuning is eliminated. We will make two clarifications. First, we will distinguish the fixed ViT image patch size from the temporal patch length used for time-series tokenization. Second, we will revise “removes the need for manual patch-size tuning” to “reduces sensitivity to temporal patch-size selection.”
>
> This claim is supported by the “PatchSize STD” results in Section 4.3, where MLLM4TS shows lower performance variation across temporal patch sizes than the time-series-only variant.
>
> ### 7. Dataset-Specific Differences
>
> We thank the reviewer for pointing out cases where MLLM4TS underperforms. The visual modality is most helpful when cross-channel visual patterns are informative and the rendered plot remains legible. It may contribute less when datasets have high channel density, weak inter-channel visual structure, or coarse-grained labels. In such cases, the numerical branch may already capture the dominant signal, leaving limited room for improvement from visualization.
>
> ### 8. Justification for 50 Plotted Channels
>
> We agree that the 50-channel visualization limit should be better justified. This number is not a tuned hyperparameter and does not remove channels from the model input. It is an implementation choice for the **visual branch only**, intended to preserve plot legibility under the input-resolution constraints of the frozen vision encoder. The raw numerical branch still retains and tokenizes all channels.
>
> For high-dimensional datasets such as ECL and Traffic, plotting all channels would create severe visual clutter. The selection-based strategy keeps representative, less-redundant channels for visualization. Table 3 supports this design: selection-based reduction outperforms both dense visualization and the engineering-based alternative on ECL and Traffic. We will emphasize that the key contribution is visual scalability, while the exact number is encoder- and resolution-dependent.
>
> We thank the reviewer again for these constructive suggestions, which improve the clarity and precision of the manuscript.
>
> [1] *Bidirectional Generative Pre-training for Improving Healthcare Time-series Representation Learning.* MLHC.

---

### Review · Reviewer_Q8rF · 2026-06-08

**Summary Of Contributions:**

This paper introduces MLLM4TS, a multimodal framework designed to enhance general time-series analysis by bridging the modality gap between continuous numerical data and discrete natural language. The core idea is to explicitly render multivariate time series as color-coded, horizontally stacked line plots (images). These images are processed by a frozen vision encoder (CLIP), and the resulting visual patches are aligned with corresponding numerical time segments using a temporal-aware alignment strategy. The fused embeddings are then processed by an LLM (GPT-2). The authors evaluate MLLM4TS across classification, anomaly detection, and forecasting, demonstrating consistent improvements over unimodal baselines like OFA.

Strengths:
- Empirical Breadth: The evaluation is extensive, spanning predictive (classification) and generative (anomaly detection, forecasting) tasks, alongside zero/few-shot scenarios.

- Clever Modality Translation: Bypassing the traditional numerical-to-text tokenization bottleneck by explicitly projecting the data into a visual domain to exploit pretrained vision-language models is a highly pragmatic and effective strategy.

- Ablation Depth: The paper provides a thorough breakdown of its design choices, successfully isolating the benefits of color-coding, layout structures, and fusion stages.


Weaknesses:
- Artificial Spatial Geometry: The vertical stacking of channels creates an artificial spatial dimension in the rendered plots. While standard self-attention seems to compensate for this, this lacks the principled structural grounding found in graph-based or explicitly physically-grounded spatial representations.
= Scalability Workarounds: For high-dimensional datasets (>300 channels), the authors rely on dropping highly correlated channels to fit visual resolution limits. This is more of an engineering stopgap than a structural solution.

**Audience:**

Yes

**Audience Explanation:**

The intersection of foundation models and time-series analysis is a highly active research area. Demonstrating that the modality gap can be effectively bridged by explicitly casting numerical data into the visual domain—thereby co-opting pretrained visual reasoning capabilities—is an insightful finding. Researchers building generalist multimodal systems or working on cross-domain representation learning will find the temporal-aware patch alignment strategy particularly relevant.

**Broader Impact Concerns:**

The authors adequately discuss the broader impact in Appendix D. While time-series forecasting in critical domains (e.g., healthcare, autonomous systems, finance) carries inherent risks if model predictions are confidently incorrect, this work focuses strictly on advancing the core methodology and representation learning. No additional ethical concerns or Broader Impact Statements are required.

**Claims And Evidence:**

Yes

**Claims Explanation:**

The empirical evidence thoroughly supports the paper's core claims. The authors do an excellent job isolating the specific components that lead to performance gains. For instance, Figure 6 effectively demonstrates that the channel-wise color coding directly contributes to the model's ability to capture cross-channel dependencies, outperforming both a plain non-colored plot and a time-series-only baseline. Furthermore, the adoption of robust, bias-resistant metrics like VUS-PR for the anomaly detection evaluations lends significant credibility to their results in that domain.

**Requested Changes:**

Critical to secure acceptance:
- Failure Modes of Channel Reduction: Please elaborate on the potential failure modes of the "selection-based" dimensionality reduction used for high-dimensional datasets like ECL and Traffic. If highly correlated channels are simply dropped to avoid visual clutter, how does the framework handle highly localized anomalies that might only occur within the dropped channels? A brief discussion on the risk of masking fine-grained details during this reduction is necessary.

To strengthen the work:
- Discussion on Spatial Geometry: The visual mapping translates distinct channels to the vertical axis, creating an artificial spatial dimension. The paper would be significantly strengthened by a deeper discussion in Section 3.2 on why the vision encoder's global self-attention successfully extracts meaningful cross-channel dependencies from this artificial geometry, perhaps contrasting it with approaches that use explicit structural priors (like scene graphs or spatial modeling).
- Clarification on Few-Shot Baselines: In the few-shot evaluation (Section 4.1.4), it would be helpful to clarify the performance of state-of-the-art non-LLM baselines on these exact same data splits. This would better isolate how much of the few-shot performance gain is attributable strictly to the multimodal LLM architecture versus the general baseline capability on that specific 10% data slice.

---

> ### Author Response · Authors · 2026-06-22
>
> We thank the reviewer for the careful and positive assessment, especially for recognizing the empirical breadth, modality-translation design, and ablation depth. We address the channel-reduction concern and the two suggested improvements below.
>
> ### 1. Failure Modes of Channel Reduction
>
> We clarify that **channel reduction is visualization-only**. As described in Section 3.3, selection-based reduction only controls which channels are rendered in the visual branch to preserve line-plot legibility. The raw numerical branch still retains and tokenizes **all channels**, so no channel is removed from the model input. Thus, an anomaly in a non-rendered channel is not discarded; it remains available through the time-series branch. We will make this guarantee explicit in the revision.
>
> The selection rule further reduces visual information loss by dropping the most redundant, highly correlated channels while keeping less-correlated ones. Thus, a dropped channel often has a retained correlated proxy in the visualization. The remaining corner case is when two globally correlated channels diverge only during rare abnormal intervals. In this case, the visual branch may miss the localized deviation, but the raw branch still preserves the complete signal. The anomaly-detection results in Table 1, where MLLM4TS consistently outperforms OFA and strong task-specific baselines, suggest that the visual branch provides useful complementary signals in most settings.
> We will revise the limitations section to present channel selection as a visualization-scalability trade-off and discuss correlation-aware grouping, anomaly-aware channel selection, and hierarchical visual summaries as future work.
>
> ### 2. Spatial Geometry Discussion
>
> We agree that the spatial geometry deserves clearer discussion and will expand Section 3.2. The vertical axis should be interpreted as a **channel index**, not a physical spatial dimension. Since the visual encoder uses ViT-style global self-attention, it is not restricted to local pixel neighborhoods: patches from distant rows and columns can interact directly. Cross-channel dependencies can therefore be inferred from visual content, such as aligned peaks, shared periodicity, synchronized changes, or co-occurring waveform patterns, rather than from a fixed spatial prior.
>
> The layout ablations in Table 4 support this view. MLLM4TS remains effective under different layouts, including grid and horizontal layouts, and the horizontal layout with temporal-aware patch alignment performs especially well. This suggests that the gain does not depend on one artificial 2D arrangement, but on preserving useful temporal and cross-channel patterns that can align with the numerical branch. We will also contrast our design with graph-based or spatial-prior methods. Such priors are useful when the structure is known and reliable, but may reduce generality when the structure is unavailable, noisy, or dataset-specific. Our design instead prioritizes generality and reuse of pretrained vision encoders, allowing global attention to infer relations from visual content.
>
> ### 3. Few-Shot Baseline Clarification
>
> To better isolate the source of the few-shot gain, we added experiments on the same 10% training splits using TimePFN [1], a strong non-LLM prior-fitted network, and OFA [2], the LLM-only baseline.
>
> | Dataset | MLLM4TS (MSE / MAE) | OFA (MSE / MAE) | TimePFN  (MSE / MAE) |
> |---|---|---|---|
> | Weather Avg | **0.236 / 0.274** | 0.238 / 0.275 | 0.266 / 0.285 |
> | ETTh1 Avg | **0.578 / 0.535** | 0.626 / 0.553 | 0.675 / 0.555 |
>
> TimePFN does not recover the same few-shot performance, indicating that the improvement is not simply due to the 10% training slice being easy for a strong non-LLM baseline. The comparison with OFA further isolates the visual modality, since OFA uses a similar LLM-based backbone but no visual branch. MLLM4TS improves over OFA, especially on ETTh1, reducing average MSE from 0.626 to 0.578. On Weather, the gap is smaller, 0.236 vs. 0.238, consistent with the manuscript. These results suggest that the visual branch contributes beyond LLM capacity alone.
>
> ### Planned Revisions
>
> We will make three revisions. First, in Section 3.3 and Limitations, we will state that channel reduction only affects visualization and that the raw branch retains all channels, while discussing residual failure modes. Second, in Section 3.2, we will expand the spatial-geometry discussion, clarify the role of global self-attention, connect it to Table 4, and contrast our design with explicit structural-prior methods. Third, in Section 4.1.4 / Table 13, we will add TimePFN as a non-LLM few-shot baseline and clarify the distinction among data-slice effects, LLM-only modeling, and the added visual modality.
>
> [1] Taga et al. “TimePFN: Effective Multivariate Time Series Forecasting with Synthetic Data.” AAAI, 2025.
>
> [2] Zhou et al. “One Fits All: Power General Time Series Analysis by Pretrained LM.” NeurIPS, 2023.